# Inference and effects of barcode multiplets in droplet-based single-cell assays

Caleb A. Lareau [1,2,3 ✉], Sai Ma [1,2,4], Fabiana M. Duarte [1,2] & Jason D. Buenrostro[1,2 ✉]

A widespread assumption for single-cell analyses specifies that one cell's nucleic acids are predominantly captured by one oligonucleotide barcode. Here, we show that ~13–21% of cell barcodes from the 10x Chromium scATAC-seq assay may have been derived from a droplet with more than one oligonucleotide sequence, which we call "barcode multiplets". We demonstrate that barcode multiplets can be derived from at least two different sources. First, we confirm that approximately 4% of droplets from the 10x platform may contain multiple beads. Additionally, we find that approximately 5% of beads may contain detectable levels of multiple oligonucleotide barcodes. We show that this artifact can confound single-cell analyses, including the interpretation of clonal diversity and proliferation of intra-tumor lymphocytes. Overall, our work provides a conceptual and computational framework to identify and assess the impacts of barcode multiplets in single-cell data.

[1] Department of Stem Cell and Regenerative Biology, Harvard University, Cambridge, MA 02138, USA. [2] Broad Institute of MIT and Harvard, Cambridge, MA 02142, USA. [3] Division of Medical Sciences, Harvard Medical School, Boston, MA 02115, USA. [4] Department of Biology, Massachusetts Institute of Technology, Cambridge, MA 02142, USA. ✉email: caleblareau@g.harvard.edu; jason_buenrostro@harvard.edu

D roplet-based partitioning systems have become an essential tool for single-cell genomics research. In contrast to plate-based single-cell assays, droplet-based methods, including scRNA-seq[1,2] and scATAC-seq[3,4] enable profiling of thousands of cells in a single experiment. The marked increase in throughput is achieved by parallel barcoding of cellular nucleic acids with beads containing high-diversity DNA barcodes. Critically, downstream computational analyses assume that one barcode sequence equates to one cell.

In this work, we provide multiple lines of evidence that indicate that cells often associate with multiple barcodes by (i) multiple beads occurring within the same droplet or (ii) heterogeneity of oligonucleotide sequences within a single bead (Fig. 1a). Here, we refer to these instances whereby multiple DNA barcodes occur within the same droplet as "barcode multiplets". We find that barcode multiplets can considerably impact single-cell analyses and demonstrate that rare cell events (e.g., the analysis of cell clones) can be particularly affected by this artifact. Further, we provide a computational solution to identify these barcode multiplets in existing single-cell datasets, particularly from the scATAC-seq platform. Finally, we provide recommendations to mitigate these biases in existing assays.

**Fig. 1 Quantification of barcode multiplets from multiple beads in 10× Chromium platform. a** Schematic of bead loading variation and phenotypic consequences. Droplets with 0 beads fail to profile nucleic acid from the loaded cell ("dropout") whereas barcode multiplets fractionate the single-cell data. Barcode multiplets can be generated by either heterogeneous barcodes on an individual bead or two or more beads loaded into the same droplet. The * indicates the bead multiplet that can be quantified via imaging. **b** Representative example of beads loaded into droplets from the 10× Chromium platform. The white box is magnified 3× for the panel on the right, revealing multiple beads loaded into droplets. Stars indicate beads (except 0) and are colored by the number of beads contained in the droplet. The image is representative of a total of 30 fields of view taken from three independent experiments. **c** Empirical quantification of number of bead barcodes based on image analysis over 3 replicates with previously published data (Zheng et al.[2]). **d** Percent of barcodes associated with multiplets under the distribution observed in **c**. Error bars represent standard error of mean over the experimental replicates. Source data are available in the Source Data file.

## Results

**Bead multiplets quantified through imaging.** While cell doublet rates are routinely quantified by species-mixing analyses, analogous multiplet rates for bead loading are scarcely discussed. Importantly, commonly used droplet-based assays (e.g., the 10× Chromium platform) leverage a close-packing ordering of beads[5] to load predominantly one bead per droplet, thus achieving a "sub-Poisson" distribution. First, we sought to test this assumption and empirically quantify bead loading within droplets. To achieve this, we loaded hydrogel training beads into droplets following recommended guidelines and imaged the resulting solution. Beads were readily visible and quantifiable per droplet (Fig. 1b; Supplementary Fig. 1a–d), enabling empirical estimates of the number of beads per droplet. A total of 3865 droplets spanning 30 total fields of view (FOV) over three experimental replicates were quantified (Table S1; see Methods section). Importantly, while the training beads largely do not differ from those used in the regular protocol, the training buffer (different from the typical reaction buffer) is required to visualize beads after loading.

On average, we found that 16.1% of droplets contained no beads, 80.0% contained exactly one bead, and 3.9% had two or more beads (Fig. 1c). These results were consistent with the previously reported results of this platform ("Zheng")[2] and confirm the sub-Poisson loading of beads into droplets (compare to Supplementary Fig. 1e for optimal Poisson loading). While the mean of the bead loading was consistent with previous reports, we note considerable run-to-run variability from our imaging replicates, ranging from 0.8 to 8.4% (Supplementary Fig. 1f). Furthermore, we noted occurrences of large droplets with multiple beads (Supplementary Fig. 1g) that likely originated from the errant merging of several individual droplets, yielding another source of potential barcode multiplets. While our imaging results indicate that the occurrence of bead multiplets likely varies between machines and individual runs, we note that the training kits used in our experiment are ultimately a proxy for the reagents used in producing single-cell data. Thus, our estimates may reflect greater variability in the bead doublet rate than what is present in many datasets. Regardless, our results suggest that multiple beads may co-occur in droplets and motivate additional computational analysis to determine potential barcode multiplets.

While our estimate of the occurrence of multiple beads in droplets confirms previous reports[2], we emphasize that this problem is exacerbated when considering potential barcodes in single-cell data. On average, we estimate that 11.4% of barcodes would represent barcode multiplets, reflecting droplets with heterogeneous oligonucleotide sequences (Fig. 1d; see Methods section). Moreover, we note that our estimate from imaging alone provides a lower-bound estimate for the true occurrence of barcode multiplets for two reasons. First, droplets with four or more beads were assigned a count of four since the exact number of beads could not be reliably determined in these instances (e.g., Supplementary Fig. 1d). Second, imaging cannot evaluate the possibility of heterogeneous beads, a second class of artifact that leads to barcode multiplets (Fig. 1a). Despite the potential for alarmingly high rates of barcode multiplets, the effect of this confounding phenomenon has not been systematically considered in single-cell analyses. Intuitively, these observed barcode multiplets fractionate data from the cell to multiple barcodes, resulting in a reduction of data per cell and the substantial overestimation of the total number of cells sequenced by artificial synthesis of barcodes reflecting the same single cell. With the potential for this artifact confirmed by imaging, we sought to further understand its properties and effects in single-cell data.

**Identifying barcode multiplets in 10× scATAC-seq with bap.** Recently, we developed a computational framework called bead-based ATAC processing (bap), which identifies instances of barcode multiplets in droplet single-cell ATAC-seq (dscATAC-seq)[3]. Critically, this approach discriminates between multiple true cells and barcode multiplets by considering the location of Tn5 insertion sites, noting that barcode multiplets would disproportionately amplify the exact fragments (Fig. 2a; Supplementary Fig. 2). In other words, our computational approach leverages the molecular diversity of Tn5 insertion sites across the genome to identify pairs of barcodes that share more insertion sites than expected and merge these corresponding barcode pairs (Fig. 2a). Previously, we utilized bap to facilitate super-loading beads into droplets to achieve a ~95% cell capture rate with a mean 2.5 beads/droplet[3]. We reasoned that bap may identify barcode multiplets in 10× scATAC-seq data.

After updating bap to facilitate processing of the 10× scATAC data (Supplementary Fig. 2; see Methods section), we conducted an initial in silico experiment to verify the applicability of our approach to 10× scATAC-seq data. Here, we combined two channels from a similar biological source (~5000 cells of peripheral blood mononuclear cells; PBMCs) and executed bap on the resulting combination (Fig. 2b; see Methods section). As any barcode pairs merged between channels would be false positives, our analysis facilitated an estimation of the false positive rate of our approach in 10× data. After executing bap with the default parameters, 1874 barcode pairs were identified as sharing a markedly high rate of shared transposition events. Specifically, 931 pairs from channel 1 (Fig. 2c) and 943 pairs from channel 2 (Fig. 2d) were identified. However, zero pairs were identified between channels (Fig. 2e), indicating a very low false positive rate for bap when applied to this assay. Moreover, the shape of the ranked-ordered barcode pair curves for the channels separately were distinct from the between-channel curve (Fig. 2c–e). Taken together, these results support the utility of bap in inferring barcode multiplets from the 10× platform.

After establishing the applicability of bap for 10× scATAC-seq data, we sought to better understand the properties of barcode multiplets, focusing on two datasets ("This Study" and "Public"; see Methods section) of ~5000 human PBMCs (Fig. 3a). Overall, we estimated the percentage of barcodes in multiplets were 13.2% (This Study; Supplementary Fig. 3a) and 17.6% (Public; Fig. 3b). These cell barcodes were identified from the high-quality, error-corrected barcode sequences from CellRanger with abundant reads in peaks. Additionally, since individual barcodes in the space of all possible barcodes are separated by a minimum Hamming distance of three in the 10× platform, the high prevalence of barcode multiplets is unlikely to be explained by sequencing errors alone. Importantly, these implicated barcodes are normally utilized in downstream analyses, including cell clustering and clonotype abundance estimates. Additional analyses suggested that a greater number of multiplets were present in the library but did not pass thresholds for reads detected due to the fractionation of data associated with these barcodes (Supplementary Fig. 3b; see Methods section).

Surprisingly, from these experiments, we observed instances in both datasets where barcode multiplets contained at least seven distinct barcodes (Table S2). In particular, we observed two instances of multiplets containing nine unique barcodes in the Public dataset. Of note, each implicated barcode contained a restricted longest common subsequence (rLCS) of 9 (Fig. 3c; see Methods section). We suggest that these barcode multiplets likely reflect error during barcode synthesis resulting in a single bead with multiple barcodes, resulting in a "complex bead" (Fig. 1a). Visualization of these barcode multiplets from dimensionality reduction using t-distributed stochastic neighbor embedding (t-

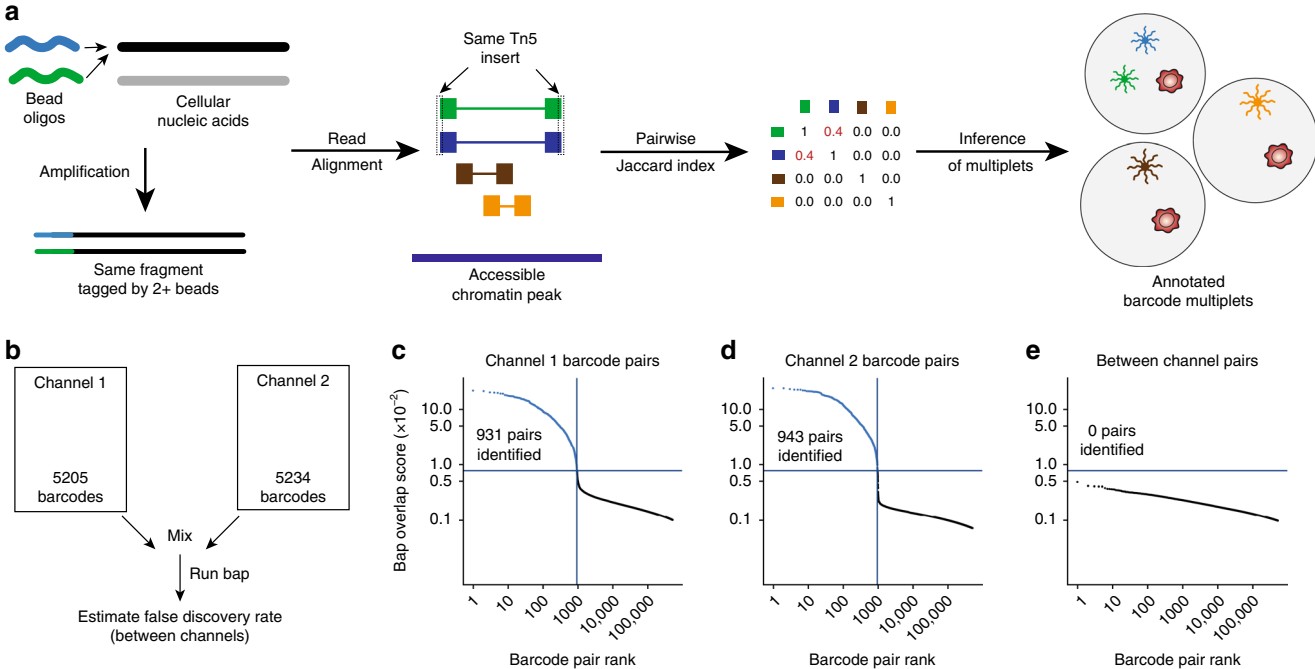

**Fig. 2 Verification of bap to identify barcode multiplets using 10× scATAC-seq data. a** Schematics of methodology to detect barcode multiplets whereby cellular nucleic acids are tagged by two different oligonucleotide sequences and later inferred from sequencing a scATAC-seq library from the same Tn5 insertions per fragment. **b** Schematic of mixing experiment. Two channels were combined and the resulting merged files were analyzed with bap. **c–e** Knee plots comparing the top 500,000 barcode pairs from **c** only channel 1, **d** only channel 2, and **e** between channels. The number of pairs calls is indicated by the number of points above the blue horizontal line (see Methods section).

SNE) confirmed these barcodes reflect markedly similar chromatin accessibility profiles (Fig. 3d; Supplementary Fig. 3c). Overall, barcode multiplets generally co-localized with barcode singlets and do not dramatically alter the interpretation of cell types in an embedding (Fig. 3e). However, we find that certain regions of the t-SNE embedding contained a disproportionate concentration of barcode multiplets, which may lead to errant identification of presumed rare cell types (e.g., five unique multiplets shown in Fig. 3e).

To further elucidate these identified barcode multiplets, we annotated these barcodes with graph-based Louvain clusters (produced using the default CellRanger execution). As expected, we observed a significant enrichment of barcode multiplet pairs occurring in the same cluster (91.1% for This Study; 74.1% for Public) compared to a permuted background (11.6% and 8.6%, respectively; Supplementary Fig. 3d; see Methods section). We note that barcode multiplets not within the same cluster largely reflect barcodes split between multiple clusters of the same cell type (e.g., myeloid cells; see Multiplet 5 in Supplementary Fig. 3c and Table S3). Additionally, we observed a statistically significant association between the Louvain cluster assignment and inferred barcode multiplet status for both This Study ($p = 0.0065$; chi-squared test) and Public datasets ($p = 2.46\text{e}{-}05$; chi-squared test; see Methods section). These results indicate that the barcode multiplets can occur in clusters unevenly, potentially confounding inferences regarding cell-type abundance. Additionally, through iteratively downsampling and re-executing bap, we confirmed the stability of our metric with sequencing depths as low as a median 10,000 fragments detected per barcode (Supplementary Fig. 3e; see Methods section), confirming the robust utility of this approach. Overall, as these barcode multiplets represent quasi-independent observations of the accessible chromatin landscape of the same single cell, we suggest that these identified barcode multiplets may be utilized in a variety of different useful applications. Examples include determining sequencing

saturation, inferring sequencing biases, and benchmarking bioinformatic clustering approaches. Furthermore, these barcode multiplets can be merged to improve data quality[3].

**Contributions of types of barcode multiplets.** Having verified the overall detection of the effects of barcode multiplets in these datasets, we sought to determine the relative contributions of each source of barcode multiplets to the overall abundance (Fig. 1a). To achieve this, we established a null distribution by computing the rLCS for random pairs of barcodes from the 10× whitelist (see Methods section). Over 1,000,000 sampled pairs, we determined that pairs with an rLCS ≥ 6 were extremely uncommon assuming an independent co-occurrence (<0.5% probability of co-occurring; Supplementary Fig. 3f). Thus, for inferred multiplets with a mean rLCS ≥ 6, we interpret these to be most likely caused by heterogeneous barcodes within a single bead. After computing the mean rLCS between pairs of barcodes per multiplet, we determined that 87.5% of multiplets were likely caused by these complex or heterogeneous beads in the Public dataset (Fig. 3f). Using this classification, we could further estimate the prevalence of these complex beads to be 6.41% in this dataset (see Methods section). Parallel analyses for This Study dataset yielded similar results (83.5% of barcode multiplets were due to complex beads; 4.95% of beads were heterogenous beads). Interestingly, the percent difference between the log2 number of valid fragments for these two classes of multiplets showed greater variability in the number of fragments per barcode for the complex beads than for barcode multiplets presumably caused by two beads (Fig. 3g; see Methods section). This result supports the idea that there may be a predominant individual barcode sequence on these complex beads though there is detectable heterogeneity. Finally, as 10× recently released their v1.1 "NextGem" design, we processed two additional datasets that were run with the two different chip designs in parallel. Our results confirm that the abundance of barcode multiplets persists across both of these two different chip

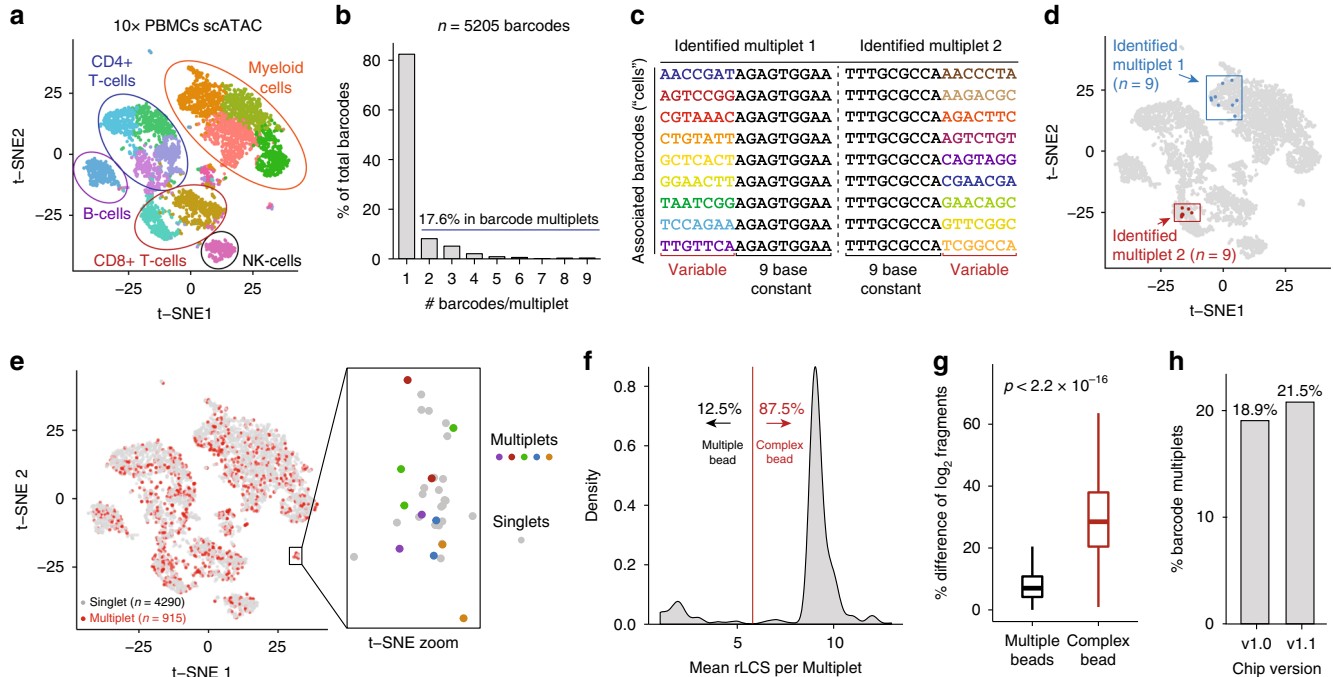

**Fig. 3 Inference and effect of barcode multiplets in single-cell ATAC-seq data. a** Default t-SNE depiction of public scATAC-seq PBMC 5k dataset. Colors represent cluster annotations from the automated CellRanger output. **b** Quantification of barcodes affected by barcode multiplets for the same dataset (identified by bap). **c** Depiction of two multiplets each composed of 9 oligonucleotide barcodes. Barcodes in each multiplet share a long common subsequence, denoted in black. **d** Visualization of two barcode multiplets from **c** in t-SNE coordinates. **e** Visualization of all implicated barcode multiplets from this dataset. The zoomed panel shows a small group of cells affected by five multiplets, indicated by color. **f** Empirical distribution of the mean restricted longest common subsequence (rLCS) per multiplet. A cutoff of 6 was used to determine either of the two classes of barcode multiplets. **g** Percent difference of the mean log2 fragments between pairs of barcodes within a multiplet. The reported p-value is from a two-sided Kolmogorov–Smirnov test. The exact *p*-value is lower than machine precision. Analysis represents $n = 5205$ barcodes over 1 experimental replicate. Boxplots: center line, median; box limits, first and third quartiles; whiskers, 1.5× interquartile range. **h** Overall rates of barcode multiplets from additional scATAC-seq data comparing v1.0 and v1.1 (NextGEM) chip designs. Source data are available in the Source Data file.

designs (Fig. 3f) as well as the rates of complex beads and multiple beads underlying the multiplets (Supplementary Fig. 3g).

**External corroboration of barcode multiplets.** In response to a pre-print version of our article[6], 10× Genomics released a letter with a software solution to identify multiplets from the output of the CellRanger-ATAC pipeline. In principle, their approach similarly utilizes the molecular diversity of Tn5 cut sites to identify putative barcode multiplets. After obtaining this script, we evaluated our two well-characterized PBMC datasets and determined that the rates of barcode multiplets were extremely similar as >98% of barcodes were concordantly classified as belonging to a barcode multiplet or not (Supplementary Fig. 3h; see Methods section). As a solution to the barcode multiplet artifact, the 10× method discards the lower abundance barcodes per multiplet. While further analysis is required to determine the optimal strategy for handling barcode multiplets, these results corroborate our estimates inferred and reported from bap.

**Confounding of clonal lymphocytes due to barcode multiplets.** We suggest that many applications of the 10× Chromium platform are unlikely to be impacted by bead multiplets. However, droplet single-cell approaches are now employed for purposes requiring increasingly precise quantitation, such as highly multiplexed perturbations[7], clonal lymphocyte analyses[8], or diagnostics[9]. Thus, for analyses of rare events, such as those routinely quantified in CRISPR perturbations or in clonal analyses of cells, the surprisingly high prevalence of barcode multiplets may

become particularly problematic. As one example, we hypothesized that barcode multiplets may significantly alter quantitation of cell clones distinguished by unique B-cell receptor (BCR) and T-cell receptor (TCR) sequences in a tumor microenvironment (Fig. 4a). Though there is no current approach to define bead multiplets in scRNA-seq data, we reasoned that certain abundant BCR and TCR clonotypes may be explained by complex beads representing one true cell (similar to Fig. 3c). To test this, we reanalyzed a publicly available dataset generated using the 10× V(D)J platform that analyzed lymphocytes from a non-small-cell lung carcinoma (NSCLC) tumor (Fig. 4a). Indeed, we observed two instances of a BCR clone with four or more cells that could be more parsimoniously interpreted as barcode multiplets derived from a single B-cell (Fig. 4b). In particular, all presumed cells from these clones shared an rLCS of ≥9, an extremely unlikely event assuming true clonal cells would be randomly assigned barcode sequences (Supplementary Figs. 3f, 4a). Indeed, the distribution of the rLCS across all BCR clonotypes indicated a detectable bias indicative of barcode multiplets (Supplementary Fig. 4a; see Methods section). Furthermore, we identified additional clones that were depicted with a more complex heterogeneous structure that still broadly reflected bead synthesis errors (Fig. 4c).

Having established the clear possibility of barcode multiplets occurring in these data, we sought to determine how the interpretation of the overall clonality would be changed when accounting for the barcode multiplets. Using conservative estimates of barcode multiplets from the scATAC-seq analyses, we conducted a series of simulations (see Methods section;

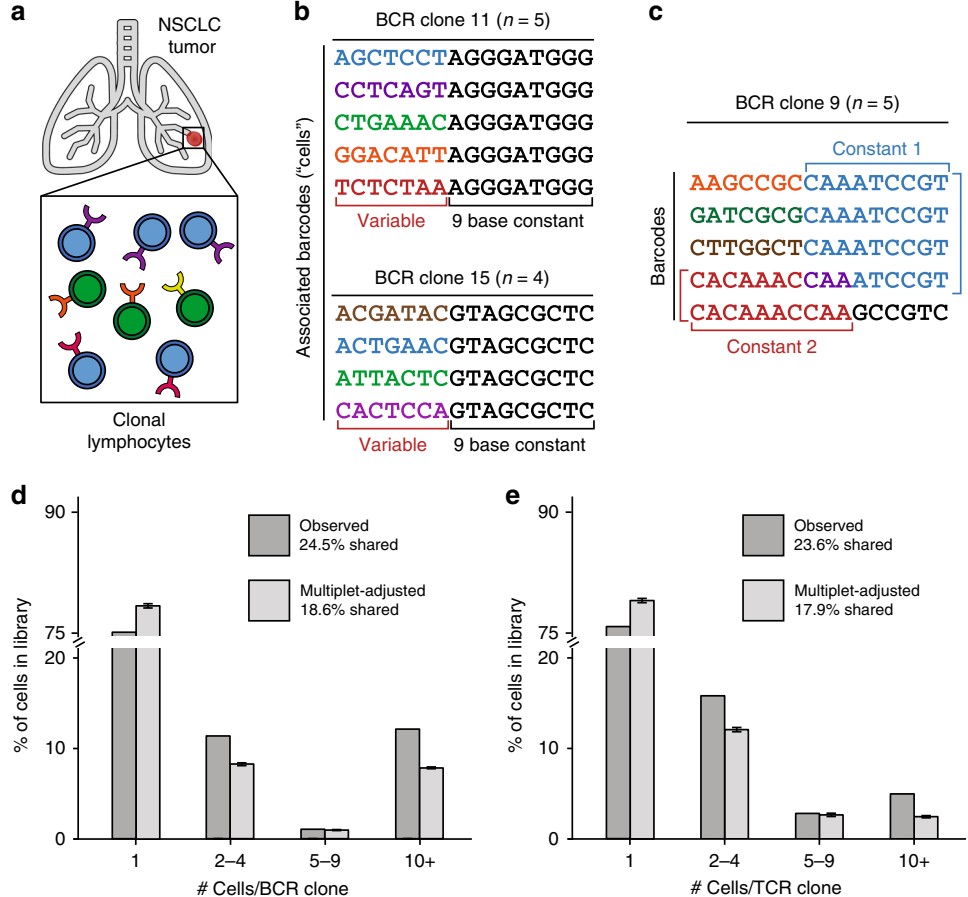

**Fig. 4 Confounding of intratumor clonal lymphocytes inference from barcode multiplets. a** Schematic of intra-tumor lymphocytes identified from single-cell V(D)J sequencing on the 10× platform. **b** Identification of two presumed clonotypes composed of five and four barcodes. These clonotypes are likely to have been derived from one cell observed multiple times via barcode multiplets. **c** Example of a presumed clone composed of five barcodes with multiple constant sequences. **d**, **e** Overall summary of prevalence of **d** B-cell and **e** T-cell clone size before and after adjusting for observed rates of barcode multiplets in single-cell data. Error bars represent standard errors of the mean across $n = 100$ independent permutations from one experimental dataset per receptor sequence. Source data are available in the Source Data file.

Table S3). Overall, the percentage of cells associated with a clonotype comprised of at least two cells decreases considerably for both BCR (24.5–18.6%; Fig. 4d) and TCR (23.6–17.9%; Fig. 4e) clonotypes. Further analyses indicated a clone false discovery rate as high as 23.5% (BCR) and 22.5% (TCR) in these data (see Methods section), painting a much more conservative picture of clonality within NSCLC tumors. The results from these simulations indicate that bead multiplets may significantly confound clonal analysis and that this quantifiable discrepancy may falsely lead to conclusions of clonal expansion of lymphocytes in primary tumors.

## Discussion

Overall, our work provides a perspective to consider barcode multiplets in single-cell data. Though the exact chemistry of the training beads and reaction is different than what is typically employed in the 10× single-cell reactions, our imaging results confirm detectable bead multiplets as previously reported[2]. Additionally, we show that bap, a computational algorithm designed to infer barcode multiplets, can be applied to sequenced scATAC-seq data from the 10× platform and confidently identify barcode multiplets. As the rates inferred from imaging and from bap are derived from distinct sources (i.e., bead/droplet counting versus sequencing), discretion is required when comparing between the detection modalities. Further analyses of multiplets

identified by bap indicate that putative heterogeneity of beads in the 10× reaction is the predominant driver of the surprisingly high rates of multiplets in these datasets. Our analyses of clonal cells marked by BCRs and TCRs further suggest that bead sequence heterogeneity may be an artifact present across multiple sources of 10× single-cell data.

Conceptually, the presence of heterogeneity in beads is unlikely to be caused by an on/off process and instead likely exists as a spectrum across all beads used in these assays. As the estimated number of complex beads relies on sufficient amplification and detection of lower-frequency barcodes inside of droplets, the proportion of barcodes affected by this artifact becomes a function of the read depth (Supplementary Fig. 3e) and the barcode threshold (Supplementary Fig. 3b), which are in turn functions of the underlying chemistry of the assays. While our estimation of the clone false discovery rate assumed comparable rates for barcode multiplets for scATAC-seq and scRNA-seq methods, technical differences across these assays could also result variable barcode multiplet abundances. As such, our work motivates further investigation into the relationship between barcode multiplets and clonal diversity across various technical platforms.

As single-cell approaches move toward the precise quantification of rare cell types, trajectories, perturbations, and clones, an understanding of potential artifacts is essential as their confounding effects may become exacerbated in large datasets.

Additionally, as these measurements move toward clinical applications[9], particularly in tumors where TCR repertoire may serve as a prognostic biomarker[10], barcode multiplets may significantly confound interpretation. In some analyses (with <15% clones), we anticipate that many identified clonal cells may arise from bead multiplets. While our existing computational approach (bap) can facilitate the identification of barcode multiplets in scATAC-seq data, further experimental and computational tools are needed to more broadly identify these effects in RNA or genome sequencing droplet-based assays. We envision a combination of dense exogenous barcodes via cell hashing[11] and evolved by CRISPR-Cas9[12] or intrinsic features such as clonal mutations, rearrangements, or highly correlated abundances with barcode sequence similarity metrics could be leveraged to better infer barcode multiplets. Such approaches would complement existing tools that robustly identify cell doublets[13,14] and empty droplets[15] from droplet-based scRNA-seq and further mitigate hidden confounders in single-cell data. Until then, we suggest that inferences regarding rare cell events should be corroborated across multiple channels or technologies to validate interpretation. Moreover, we acknowledge contexts where multiplets can be used to benchmark features of droplet-based assays as have been recently described[16].

Taken together, our estimation and identification of barcode multiplets has a wide range of potential applications and confounding effects that influence widely used droplet-based single-cell assays.

## Methods

**Loading and visualizing bead loading in droplets**. We used the 10× Chromium Controller Training Kit (PN-12024, PN-120238) to generate GEMs following manufacturer's instructions. The GEMs were carefully collected without disrupting the emulsion. After GEM formation, 10 μL of GEMs from each 10× channel was immediately loaded onto Countess Cell Counting Chamber Slides (C10228, Thermofisher) for visualization. We captured ten bright field images under an Olympus IX70 microscope, and beads per droplet were counted based on manual inspection of images. To quantify the proportion of barcodes affected by multiple beads (barcode multiplets), we used the following equation:

$$\% \text{ Multiplets} = \sum_{(b=2)}^{4} b\, n_\text{b} \Big/ \sum_{(b=1)}^{4} b\, n_\text{b} \times 100,$$

where $b$ is the number of beads present in a given droplet and $n_\text{b}$ is the number of droplets with beads. Here, the expression is capped at 4 as droplets with 4+ beads could not be reliably quantified. Thus, in these instances, the value of barcodes per droplet were conservatively assigned a count of 4. For the Zheng et al.[2] data, we used the following abundances from previous imaging data: 15% of droplets had 0 beads; 80% of droplets had 1 bead; and 5% of droplets had 2 beads. As neither the raw data nor the quantification values have been published, these values were approximated from an examination of a plot previously reported[2].

**Profiling PBMCs using 10× scATAC-seq**. For 10× scATAC-seq experiments with PBMCs (PB003F, Allcells), frozen cells were quickly thawed in a 37 °C water bath for about 30 s and transferred to a 15 mL tube. Five milliliter of pre-warmed RPMI 1640 (ATCC, 30-2001) supplemented with 10% fetal bovine serum (FBS) were added to the sample drop by drop. The cells were pelleted by spinning at 300 g for 5 min at room temperature. The supernatant was removed, and cells were washed with 1 mL PBS. The cells were then pelleted again, resuspended in 1 mL PBS, and used for 10× ATAC v1.0 protocol following manufacturer's instructions. The corresponding library was sequenced on an Illumina NextSeq 500.

**Data preprocessing**. Raw sequencing data was processed with Cell Ranger ATAC version 1.0.0. Reads were aligned to the hg19 reference genome available on the 10× Genomics website. Processed 10× PBMC datasets were downloaded from https://www.10xgenomics.com/resources/datasets/ from the version 1.1 PBMC 5k scATAC-seq dataset. The requisite input files for bap included the.bam file and the high-quality barcodes file. Additional annotations from Louvain clustering and t-SNE coordinates were also downloaded for downstream visualization and analyses. For the comparison of the chip technologies (Fig. 3g), we again downloaded the PBMC 5k scATAC-seq datasets from the "Chromium Next GEM ATAC Demonstration."

**Processing 10× scATAC-seq data with bap**. In order to facilitate the processing of 10× scATAC-seq data with bap, no major substantive changes were required for the underlying barcode multiplet identification algorithm that has been previously outlined[3]. However, additional command-line options were added, including the—barcode-whitelist flag, which imports the error-corrected, quality-controlled barcodes identified as "cells" by CellRanger, enabling analysis of the filtered output from the default 10× pipeline. This functionality augments the default process in bap where abundant barcodes are identified via quantification and knee-calling in terms of total reads observed per barcode. Versions 0.5.9+ of bap facilitate full analysis and merging of barcode multiplets with 10× scATAC-seq data.

**In silico mixing experiment**. Using two different public PBMC 5k datasets, we sought to determine a putative false positive rate for the application of bap to 10× scATAC-seq data. Here, we denoted the PBMC-5k "Public" dataset as Channel 1 and the PBMC-5k from the NextGEM beads as Channel 2. We modified the CB tags (which contains the error-corrected barcodes) in the.bam files for each channel to ensure that each barcode for each experiment was uniquely identifiable. These modified bam files were subsequently merged. Next, the same modification to the barcodes was made, and the two high-quality barcodes files were combined into a single file. We then executed bap using the default parameters with this merged.bam and merged barcode list file. Using a single threshold determined by the knee call, we identified pairs of barcodes originating from the same or different channels as summarized in Fig. 2c–e. The top 500,000 barcode pairs were plotted in rank order for each of these three plots, and the same single threshold was visualized in all three panels.

**Assigning bead barcodes to multiplets**. The identification of multiplets follows the same strategy previously described[3]. In brief, a per-barcode pair summary statistic (modified jaccard index) is computed using the one base pair location of Tn5 insertions. We emphasize that this statistic has been validated using an orthogonal oligonucleotide library as we have previously described[3]. From this distribution of millions of barcode pairs, we computationally infer an inflection point threshold $T$ (similar to a"knee-call" used by CellRanger to identify true cell barcodes). To derive multiplets, we iteratively consider the barcode pairs (e.g., $b_1$ and $b_2$) with the highest remaining overlap score and append any additional barcodes whose overlap value with either $b_1$ or $b_2$ exceeds $T$. For example, if the statistic between $b_1$ and $b_3$ exceeds $T$, then $b_1$, $b_2$, and $b_3$ are assigned to one multiplet. This process continues until all barcodes are assigned a multiplet that had an overlap score exceeding $T$. All remaining barcodes are assigned as singlets. To facilitate processing of the 10× scATAC-seq data, we modified the command line interface and internal data structures of bap, but the conceptual basis and execution is the same as previously described[3].

**Classifying and quantifying complex beads**. To determine multiplets driven by putative bead barcode synthesis errors, we considered all pairs of barcodes within an annotated multiplet and computed the restricted longest common subsequence (rLCS) between them. Explicitly, the rLCS is the largest consecutive number of characters that match between two strings without shifting the strings. We note the necessity of defining a distance metric (rLCS) that is distinguished from the longest common subsequence (LCS) as our metric does not allow insertions or deletions when performing the string matching. Additionally, rLCS is distinguished from the Hamming distance as the matching characters must all occur in a continuous unit (which is not enforced by Hamming).

To determine an appropriate threshold to classify multiplets as having originated from multiple beads or a single heterogeneous bead, we established a null distribution of the rLCS shown in Supplementary Fig. 3f. To achieve this, 1,000,000 random draws of barcode pairs were determined and the rLCS was computed. We selected an rLCS threshold of 6 as pairs with an rLCS ≥6 represented less than 0.5% of the data, which was used to classify multiplets from the real data (Fig. 3f). To determine whether the number of fragments was similarly captured between barcodes contained in multiplets, we computed the pairwise percent difference of the log2 unique fragments ("passed_filter" in the CellRanger-ATAC.csv file). The per-multiplet average of the mean pairwise percent difference is plotted in the boxplots in Fig. 3g, and we used a two-sided Kolmogorov–Smirnov test to verify that the droplets containing multiple beads had a more even ratio of reads compared to multiplets driven by bead heterogeneity.

To quantify the percent of beads that had heterogeneity, the numerator was the number of multiplets identified with an rLCS ≥ 6 (from Fig. 3f). The denominator was the total number of barcodes analyzed while (1) still counting all barcodes in perceived bead multiplets but (2) collapsing the heterogenous barcode multiplets to only 1 barcode. For example, in the "This Study" dataset, the total number of barcodes passing the CellRanger knee was 5453. Of these, 4732 barcodes were from singlets, 121 barcodes were associated with multiplet beads per droplet (and thus not complex), and 600 barcodes were associated with 253 complex beads. The complex bead rate can be computed as follows:

$$\text{Complex bead rate} = \frac{\# \text{ complex beads}}{\# \text{ singlet beads} + \# \text{ beads in bead multiplets} + \# \text{ complex beads}}$$

For our example of the "This Study" dataset:

$$\frac{253}{4732 + 121 + 253} = 4.95\%$$

**Chi-squared test for cluster/multiplet**. To test for association between barcode multiplets and cluster identification, we performed a chi-squared test for independence. For the $n$ Louvian clusters identified by CellRanger, we assembled a $2 \times n$ contingency table, tabulating barcodes into corresponding entries in the contingency table. The two rows specified whether each bead barcode was predicted to occur in a multiplet or not as identified by bap. $P$-values were computed using the chi-squared statistic with $n - 1$ degrees of freedom.

**Evaluation of bap with variable input barcodes**. To test the abundance of barcode multiplets with different numbers of considered barcodes, we executed bap with 5000–10,000 barcodes at intervals of 1000 barcodes (six additional executions) in addition to the 5205 found by CellRanger's knee call. Each barcode set was nominated based on the ranking of fragments in peaks, the same metric used by CellRanger to determine an optimal threshold. Our results (Supplementary Fig. 3b) show that the inferred cutoff underestimates the barcode multiplets in the Public data, consistent with our imaging results. We interpret this plot to show that barcode multiplets often occur near the inflection point (consistent with these barcodes having fewer reads due to the fractionated data). However, this rate flattens when additional barcodes added do not represent multiplets but other ambient fragments that cannot be associated with a highly-observed barcode.

**Enrichment for barcode multiplet pairs in the same cluster**. For each barcode multiplet identified by bap, we considered all possible pairwise combinations of constitutive barcodes. For example, multiplets consisting of precisely two bead barcodes had one pair whereas multiplets consisting of four barcodes contained six barcode pairs (all combinations; choose two). For these pairs, we computed the proportion that occurred in the same Louvian cluster produced by the default CellRanger execution. A background rate was generated by performing 100 permutations of the full dataset where cluster labels were permuted.

**Downsampling analyses**. To evaluate the stability of the bap statistic as a function of coverage, we downsampled the dataset generated here ("This Study") at intervals of 10% and reran bap on the resulting downsampled.bam files. Here, we used the full set of high-quality barcodes determined from the CellRanger execution on the full dataset. Moreover, we determined the set of identified barcode pairs from the full dataset as a "true positive" set of pairs to compare the downsampled results. Supplementary Fig. 3e shows the results of this downsampling, including the 40% subsample (that corresponded to a median 10,132 fragments per barcode) that achieved >90% sensitivity in detecting the set of barcode pairs from the full data. Critically, in each of the nine downsampled executions of bap, no barcode pairs were identified that were not present in the full dataset.

**Comparison with 10× solution**. After contacting 10× support, we obtained the "clean_barcode_multiplets_1.0.py" script, which identifies barcode multiplets in single-cell ATAC-seq data. We executed this code and evaluated the output for the two scATAC-seq datasets closely analyzed in this work ("Public" and "This Study"). While the procedure used to identify multiplets similarly utilizes shared Tn5 insertions, the treatment of multiplets once detected is different from bap. Specifically, for each multiplet, the barcode with the most unique fragments is retained and the other barcodes are filtered out. Further, 10× refers only to the barcodes that are filtered out as "multiplets", rather than counting the most prevalent barcode as part of a barcode multiplet as we've done throughout this manuscript. For comparison purposes, we used our definition of barcode multiplet (as stated in the abstract) and reported the rates from each tool (see script in Code Availability for the exact procedure). Finally, to compute the concordance between the two methods, we assigned each barcode whether or not it was part of a barcode multiplet from both sources and report the percentage of barcodes that had a matching annotation across the detection methods.

**Estimation of multiplet-adjust BCR/TCR clonotype abundances**. In order to estimate the number of cells contributing to each clonotype (defined by a unique BCR or TCR sequence), we downloaded the per-barcode clone identification files (BCR: vdj_v1_hs_nsclc_b_all_contig_annotations.csv; TCR: vdj_v1_hs_nsclc_t_clonotypes.csv) from the 10× CellRanger output for the public NSCLC tumor dataset. Here, each barcode is assigned a clonotype group when detected with high confidence in the CellRanger pipeline. To simulate the occurrence of barcode multiplets, we executed the following simulation procedure.

For each barcode $i$ with a total of $n$ barcodes in the experiment (all assigned a clonotype), we simulate a corresponding multiplet value $m_i$ which defines the barcode multiplicity; i.e., the number of unique barcodes that overall co-occur with barcode $i$ inside a theoretical droplet. We performed our simulation by specifying the following probability distribution function:

$$P(m_i = 1) = 0.93; \; P(m_i = 2) = 0.05; \; P(m_i = 3) = 0.01;$$
$$P(m_i = 4) = 0.005; \; P(m_i = 5) = 0.005$$

Importantly, the values defined in the probability distribution function are grounded in the empirical estimates from bap across our two datasets but likely represent conservative estimates assuming a similar distribution of barcode multiplets from scATAC-seq holds in this assay (see Table S3). In other words, $P(m_i = 1) = 0.93$ is likely overestimated and $P(m_i > 5) = 0$ is underestimated, and from this parameterization, the expected rate of barcode multiplets is 15.8%. Here, we denote the set of values $m_i$ as $M$ (of length $n$). To account for $k$ clonotypes with exactly one barcode that could only be generated from a barcode singlet, we define a new set $M'$ such that $M' \cup K = M$ where $|K| = k$ and $\forall m_i \in K$, $m_i = 1$. Thus, the elements of $M'$ represent the barcode multiplicities for clonotypes annotated with two or more cells.

To estimate the multiplet-adjusted cell number per clonotype, we iteratively sample from the set $M'$ until we have observed sufficient barcode numbers to explain the original clonotype abundances, akin to observing droplets with variable barcode abundances. More precisely, for a given clonotype $j$ comprised of $c_j$ barcodes (from the raw CellRanger output), we seek to compute the multiplet-adjusted number of cells $c'_j$. To achieve this, we sample from $M'$ until the sum meets or exceeds $c_j$; $c'_j$ then is the number of draws corresponding to the number of multiplet-aware droplets needed to explain the clonotype abundance and can be interpreted as the number of cells present in the clone under the simulation setting. As an example, suppose $c_j = 4$, representing a clone of four barcodes. If we sample a 4 or 5 from $M'$, then $c'_j = 1$, meaning that one droplet explains the clone in this scenario. Last, the new per-clonotype abundances in the library are then represented by the union of $K$ with the set of all $c_j$. These multiplet-adjusted abundances were computed over 100 iterations, and the numbers reported in the main text represent the mean over these simulations. We note that an R script that achieves this approach is available in the repository noted in Code availability.

We define the "clone false discovery rate" as the proportion of clonotypes with at least two cells that then becomes explained by a barcode multiplet (i.e., $c'_j = 1$; $c_j > 1$) under our simulation setting. The numbers reported in the main text represent means for each of the BCR and TCR clones over the 100 simulations. Finally, we note that while this simulation assumes that the multiplet rates inferred for scATAC-seq are transferable to scRNA-seq, alternative approaches, such as estimating the complex bead rate from scRNA-seq directly, are likely unreliable without a sensitive multiplet detection approach as presented with bap. Ultimately, our simulation results provide an anchor to interpret the potential shift in clonotype abundance from the lens of our barcode multiplet artifact. However, additional experiments and analytical tools are needed to accurately determine clonotype abundance.

**Determination of multiplet-driven clonotypes**. In scATAC-seq data, barcode multiplets were identified using our approach previously described. However, no such approach exists for scRNA-seq. Thus, to identify potential multiplets, we were required to consider potential multiplets defined only by barcode similarity, which would be reflective of synthesis errors resulting in a bead with heterogeneous barcodes (Fig. 1a). To determine these potential multiplets, we considered all pairs of barcodes within an annotated clonotype and computed the restricted longest common subsequence (rLCS) between them. Analysis of the distribution of pairs (Supplementary Fig. 4a) within clonotype labels revealed was used to identify the clones shown in Fig. 4. When computing a permuted distribution (Supplementary Fig. 4a), labels of clonotypes were shuffled such that random barcode pairs were considered.

**Reporting summary**. Further information on research design is available in the Nature Research Reporting Summary linked to this article.

## Data availability

The public 10× scATAC-seq datasets are available for download at https://support.10xgenomics.com/single-cell-atac/datasets and the public NSCLC clontypes at https://www.10xgenomics.com/solutions/vdj/. Sequencing data generated as part of this work is available at the Gene Expression Omnibus under accession GSE143197. All other data are available from the authors upon reasonable request. Source Data are available in the Source Data file.

## Code availability

Software associated with the barcode multiplet identification and merging algorithm is available at https://github.com/caleblareau/bap. Code and data to reproduce the main findings of this study are available at https://github.com/caleblareau/barcode-multiplets.

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

## Acknowledgements

We thank J. Ulirsch and members of the Buenrostro lab for insightful comments. We are grateful to A. Labade and L. Ludwig for technical assistance. We thank Z. Burkett and R. Lebowsky of Bio-Rad for helpful conversations. We acknowledge a useful blog post from L. Pachter discussing sub-Poisson bead loading. J.D.B., C.A.L., S.M., and F.M.D. acknowledge support by the Allen Distinguished Investigator Program through the Paul G. Allen Frontiers Group. This work was further supported by the Chan Zuckerberg Initiative. C.A.L. is supported by F31 CA232670 from the NIH.

## Author contributions

C.A.L. and J.D.B. conceived and designed the study. C.A.L. implemented the software and performed analyses. S.M. and F.M.D. performed experiments and aided analyses. J.D.B. supervised the work. All authors participated in the writing of the manuscript.

## Competing interests

The authors declare the following competing interests: J.D.B. holds patents related to ATAC-seq. All other authors declare no competing interests.
