## [Peer Review File · Nature Communications]

Reviewers' Comments:

Reviewer #1:

Remarks to the Author:

Lareau et al highlighted an often overlooked artefact in the analysis of droplet-based single cell sequencing data, where a single droplet/cell may be associated with multiple droplet barcodes (multiplets). Focusing on the 10x Chromium platform, the authors estimated both empirically and computationally the rate of such multiplets and the proportion of barcodes in multiplets, which appeared to be considerably higher than previously reported (Zheng et al 2017). They further demonstrated that applications concerning quantification of rare cellular phenotypes/events are particularly affected.

The subject of this manuscript is of great interest to the single-cell genomics community. The experiment and analysis are well designed, undertaken and presented. Below are a few points on which the manuscript could potentially improve (particularly 1 & 2):

1) The empirical estimate of the proportion of empty droplet, singlets and multiplets differ considerably from Zheng et al (fig1c). As both sets of numbers are based on imaging of droplets, could the authors speculate the sources of such large discrepancy? Despite the beads per droplet distribution being more spread out than previously reported or what was reported for close-pack ordered loading, it is still far tighter than optimal Poisson loading, which makes it puzzling that the proportion of droplets with 4+ beads far exceeds those with 3 beads, a trend not seen even in Poisson loading (figS1e). One possible way to support this observation might be to show bap-derived distribution of barcodes per droplet (similar to fig2b) after discounting droplets with barcodes sharing long rLCS at relaxed cell calling threshold. Multiplets with high bead counts are likely filtered out with stringent threshold, which might explain the lack of support of the above trend from fig2b.

2) The computational method, bap, was developed and verified for a different droplet platform with higher read counts per droplet than 10X and under a different bead loading strategy. Although the principle of finding barcode pairs from the same droplet remains the same, it would be good to perform a verification experiment with random oligonucleotide as did before, or at the very least show the distribution of jaccard index/bap score for all pairs of barcodes.

3) The finding of long rLCS shared between putative multiplet barcodes is very interesting. Though not essential, it would be of great interest to know the prevalence of this seemingly synthesis error. It appears to be rare but the power of detection is probably low considering that relatively high "multiplex" is needed for confidence and each individual barcode must have sufficient reads to pass filter. Again, a verification experiment with random oligonucleotide might help here.

Reviewer #2:

Remarks to the Author:

10X Chromium has emerged as the leading platform for various types of single cell genomics. At the core of all methods is that one cell and one bead carrying one barcode sequence end up in one

reaction droplet, so that all reads resulting from one cell are identifiable by that barcode. Here the authors report, that reads from one cell can be split between two or more barcodes. This could be due to (I) multiple beads per droplet and (II) one bead carrying multiple barcodes. The first option is evaluated using microscopy of the droplets, where the authors find with more than 20% an unusually high fraction of empty droplets and 8% droplets with multiple beads, resulting in 27% of all barcodes to represent cells with barcode multipliers. The second option is evaluated by analysing the Tn5-cut site similarity in scATAC-seq data, using their published software bap that was designed to distinguish multiple cells that were caught in the same droplet. Here the authors turn this logic around and use bap to identify cells with unusually similar cut-sites, reasoning that those barcodes mark split cells as they would result from multi-barcode droplets. Using this method, the authors find 13 and 18% of barcode multipliers. As the authors say bap was calibrated to find multiple cells / droplet, which is a very different problem in that it aims to find dissimilar cells. The authors did not show anywhere that it works equally well to identify the opposite end of the spectrum, i.e. to identify cells that are unusually similar. Even though these numbers appear to be underestimates, I would still be interested in the estimate of a false positive rate.

The paper aims to quantify a technical artifact, however in my opinion neither of the methods was shown to be truly quantitative.

As far as I understood the authors tested only one run from one 10X machine under standard settings for the number of beads/droplet. In order to generalise this conclusion, I would like to see replicates, maybe even on a different machine, to catch the full technical variance. This is of particular importance because the authors already report unusual behavior of their current results with respect to the fraction of empty beads. It is necessary to exclude the possibility that their observation is due to a badly calibrated 10X machine.

The entire second part is based on the premise that barcodes from the same droplet can be faithfully identified using their previously published method bap. To begin with, the method needs a more detailed description and evaluation:

What exactly went into the calculation of the Jaccard index (Coverage filters ...)?

How does the metric depend on coverage?

Could the multiple barcodes per cell be due to sequencing errors? What is the Hamming distance between the collapsed barcode pairs?

What are the ratios of reads for the split barcodes? For the 27% of split-cells that are due to multiple beads, I would expect a more even ratio than for mixed barcodes on the same bead.

What is the false positive rate for the bap-barcode merging? The authors could analyse technical replicates from 2 different 10X runs to explore how often the Jaccard index would suggest to merge cell barcodes from two distinct runs.

If part of the split-cells can be identified by a simpler measure such as the Hamming-distance between barcodes, maybe in combination with a frequency check, similar to the suggested algorithm in UMI-tools, the generalisation from ATAC-seq to other data would be more convincing. I am missing a lot of information pertaining the experimental details, i.e. which 10X chemistry was used? Did this differ between the public and their own dataset etc ... This is of utmost importance for this paper because it describes a technical artifact, therefore the scope this analysis critically depends on such technical details.

We are grateful to the reviewers for their extremely useful suggestions that have resulted in a substantially revised manuscript. In particular, we emphasize the following additions in this new version:

- We have added replicate imaging experiments to clarify the number of bead doublets. The addition of these replicates result in averages that more closely matches rates reported in Zheng *et al.* 2017, and notably highlight the degree of experiment-to-experiment variability.
- We provide additional analyses that demonstrate a very low false discovery rate for our computational tool on 10x scATAC-seq data.
- We have added analyses that establish estimates for previously unreported effects, including the percent of beads likely affected by presumed heterogeneity (~5-7%) and differences in the variability of fragment abundance detected from the types of barcode doublets. Both of these findings were driven from helpful suggestions from the reviewers.
- We confirm that barcode multiplet artifact is still present at similar levels in the NextGEM (v1.1) chips that have been recently released by 10x.

Reviewer #1 (Remarks to the Author):

Lareau et al highlighted an often overlooked artefact in the analysis of droplet-based single cell sequencing data, where a single droplet/cell may be associated with multiple droplet barcodes (multiplets). Focusing on the 10x Chromium platform, the authors estimated both empirically and computationally the rate of such multiplets and the proportion of barcodes in multiplets, which appeared to be considerably higher than previously reported (Zheng et al 2017). They further demonstrated that applications concerning quantification of rare cellular phenotypes/events are particularly affected.

The subject of this manuscript is of great interest to the single-cell genomics community. The experiment and analysis are well designed, undertaken and presented. Below are a few points on which the manuscript could potentially improve (particularly 1 & 2):

We thank the reviewer for the overall positive reception of our work.

1) The empirical estimate of the proportion of empty droplet, singlets and multiplets differ considerably from Zheng et al (fig1c). As both sets of numbers are based on imaging of droplets, could the authors speculate the sources of such large discrepancy? Despite the beads per droplet distribution being more spread out than previously reported or what was reported for close-pack ordered loading, it is still far tighter than optimal Poisson loading, which makes it puzzling that the proportion of droplets with 4+

beads far exceeds those with 3 beads, a trend not seen even in Poisson loading (figS1e). One possible way to support this observation might be to show bap-derived distribution of barcodes per droplet (similar to fig2b) after discounting droplets with barcodes sharing long rLCS at relaxed cell calling threshold. Multiplets with high bead counts are likely filtered out with stringent threshold, which might explain the lack of support of the above trend from fig2b.

We thank the reviewer for pointing out this important point and agree that the large discrepancy warranted further investigation. To these ends (and based on suggestions from the other reviewer), we've now included replicates of the imaging experiments (discussed in greater detail in response to reviewer 2), which on average more closely align with prior reports presented in the Zheng *et al.* 2017 paper. We've now significantly revised our depiction of these results (in **Figure 1c,d**; also shown below) as well as the text summary of these findings to more appropriately reflect the abundances of multiple beads in droplets.

Figure: (c) Empirical quantification of number of bead barcodes based on image analysis over 3 replicates with previously published data (Zheng *et al.* 2017²). (d) Percent of barcodes associated with multiplets under the distribution observed in (c). Error bars represent standard error of mean over the experimental replicates. Panels were taken from Figure 1.

To our knowledge, 10x has not published any more current information about this result nor raw data (as supported by this link in their Q&A): <https://kb.10xgenomics.com/hc/en-us/articles/218166923-How-often-do-multiple-Gel-Beads-end-up-in-a-partition->. Additionally, we highlight the following changes to the text that also provide a new perspective on this potential variability:

On average, we found that 16.1% of droplets contained no beads, 80.0% with exactly one bead, and 3.9% having two or more beads (**Fig. 1c**). These results were consistent with the previously reported results of this platform (“Zheng”)² and confirm the sub-Poisson loading of beads into droplets (compare to **Fig. S1e** for optimal Poisson loading). While the mean of the bead loading was consistent with previous reports, we

note considerable experiment-to-experiment variability from our imaging replicates, ranging from 0.8% to 8.4% (**Fig. S1f**). These results indicate that the occurrence of bead multiplets likely varies between machines and individual runs.

Next, the idea to examine the number of beads loading in the context of the bap result is an excellent idea. Using 1,000,000 random draws of barcode pairs from the valid 10x carcodes, we could establish a “null” distribution for barcode similarity, reasoning that ~6 was a reasonable cutoff to discriminate these two types of differences (left panel below). Next, we considered only multiplets likely to be derived from multiple beads (mean rLCS < 6; see center panel below). From here, we examined the distribution of the numbers of barcodes (or assumed beads) per droplet from bap (see right panel below). Here, the distribution looks like what a more intuitive result would resemble as the reviewer suggests (resembling something similar to Poisson decay).

Figure: (left; **Fig. S3F**) Distribution of the restricted longest common subsequence (rLCS) for 1,000,000 randomly-sampled barcode pairs in the 10x barcode university. A threshold at 6 is drawn for use in other analyses. (center; see **Fig. 3f**) Empirical distribution of the mean restricted longest common subsequence (rLCS) per multiplet. A cutoff of 6 was used to determine either of the two classes of barcode multiplets. (right) Distribution of the estimated number of beads per drop using the presumed bead multiplets.

Ultimately, after performing additional imaging and in light of these bap results, the observation of more droplets with 4+ beads than 3 beads is less convincing. However, we do acknowledge an additional panel added in **Fig. S1** that shows what appear to be many droplets merged together, creating a larger droplet with 6 beads. If the observation is in fact generally true that droplets with 4+ beads occur more frequently than droplets with 3 beads, we suggest that this could be a possible source.

Figure (see Fig S1g): Example of droplet merge resulting in multiple beads from FOV 6 of imaging round 2.

We note the discussion of this possibility in the main text:

Furthermore, we noted occurrences of large droplets with multiple beads (**Fig. S1g**) that likely originated from the errant merging of several individual droplets, yielding another source of potential barcode multiplets.

2) The computational method, bap, was developed and verified for a different droplet platform with higher read counts per droplet than 10X and under a different bead loading strategy. Although the principle of finding barcode pairs from the same droplet remains the same, it would be good to perform a verification experiment with random oligonucleotide as did before, or at the very least show the distribution of jaccard index/bap score for all pairs of Barcodes.

We agree that ensuring the accuracy of bap on the 10X scATAC-seq platform is an important verification. Consistent with the recommendation, we attempted an analogous validation random oligonucleotide experiment as we've previously demonstrated for the dscATAC-seq assay. After failed attempts, we discussed these experiments with colleagues in industry. We now better appreciate the differences in chemistry between the droplet-based scATAC assays. Notably many of the details of the platform are proprietary, and unfortunately we feel that this experiment would be prohibitively difficult to optimize and take significant time and resources to calibrate. As such, we believe this experiment may be outside the scope of this manuscript.

Nevertheless, we agree that it is essential to validate the use of bap with 10x scATAC-seq data. To achieve this, we note the inclusion of the following computational validation in the new **Figure 2** that demonstrate both the distribution of the overlap scores (as suggested by this reviewer) and a putative false-positive rate inferred by combining two experiments (as suggested by the second reviewer). The purpose of the mixture was to determine the number of barcode pairs that are implicated between experiments as these would all be false positives. Overall, these panels outline the experimental result (**Fig. 2b**) and then corroborate a clear knee (note the log-scale) that is identified by bap (**Fig. 2c,d**) that is absent from the between-channel

barcode pairs (**Fig. 2e**), verifying the utility of our computational approach when applied to 10x scATAC-seq data.

Figure (see Figure 2): (b) Schematic of mixing experiment. Two channels were combined and the resulting merged files were analyzed with bap. (c-e) Knee plots comparing the top 500,000 barcode pairs from (c) only channel 1, (d) only channel 2, and (e) between channels. The number of pairs calls is indicated by the number of points above the blue horizontal line (see Methods).

Though this mode of verification is slightly different than what was suggested by the reviewer, we believe these results to demonstrate the very high robustness of our approach and verify the use of bap in the 10x scATAC-seq platform. We highlight the corresponding edit in the text:

After updating bap to facilitate processing of the 10x scATAC data (**Fig. S2**; see **Methods**), we conducted an initial *in silico* experiment in order to verify the applicability of our approach to 10x scATAC-seq data. Here, we combined two channels from a similar biological source (~5,000 cells of peripheral blood mononuclear cells; PBMCs) and executed bap on the resulting combination (**Fig. 2b**; see **Methods**). As any barcode pairs merged between channels would be false positives, our approach facilitated an estimation of the false positive rate of our approach in 10x data. After executing bap with the default parameters, 1,874 barcode pairs were identified as sharing an unusual number of shared transposition events. Specifically, 931 pairs from channel 1 (**Fig. 2c**) and 943 pairs from channel 2 (**Fig. 2d**) were identified. However, zero pairs were identified between channels (**Fig. 2e**), indicating a very low false positive rate for bap when applied to this assay. Moreover, the shape of the ranked-ordered barcode pair curves for the channels separately were distinct from the between-channel curve (**Fig. 2c-e**). Overall, these results support the utility of bap in inferring barcode multiplets from the 10x platform.

3) The finding of long rLCS shared between putative multiplet barcodes is very interesting. Though not essential, it would be of great interest to know the prevalence of this seemingly synthesis error. It appears to be rare but the power of detection is probably low considering that relatively high "multiplex" is needed for confidence and each individual barcode must have sufficient reads to pass filter. Again, a verification experiment with random oligonucleotide might help here.

We agree that this is a very interesting point and now use the fact that we can discriminate barcode multiplets into these two categories using the mean rLCS. We established an rLCS threshold of 6 based on a simulation using the null distribution of barcode pairs (see point 1 or **Fig. S3f**). From this, we could classify multiplets, particularly those likely to have originated from shared barcode similarity (shown below and also in **Fig. 3f**) Using this result, we could perform straightforward calculation to estimate the prevalence, which was between 5 and 7% for the examined datasets (and now reported in our abstract).

Figure: (see Fig. 3f) Empirical distribution of the mean restricted longest common subsequence (rLCS) per multiplet. A cutoff of 6 was used to determine either of the two classes of barcode multiplets.

Based on this suggestion from the reviewer and the pertinence of these results, we've now added a section that is dedicated to this effect, highlighted below:

Contributions of types of barcode multiplets

Having verified the overall detection of these effects in these datasets, we sought to determine the relative contributions of each source of barcode multiplets to the overall abundance (**Fig. 1a**). To achieve this, we established a null distribution by computing the rLCS for random pairs of barcodes from the 10x whitelist (see **Methods**). Over 1,000,000 sampled pairs, we determined that pairs with an rLCS ≥ 6 were extremely uncommon assuming an independent co-occurrence ($<0.5\%$ probability of co-occurring; **Fig. S3f**). Thus, for inferred multiplets with a mean rLCS ≥ 6 , we interpret these to be most likely caused by heterogeneous barcodes within a single bead. After computing the mean rLCS between pairs of barcodes per multiplet, we determined that 87.5% of multiplets were likely caused by these complex or heterogeneous beads in the public dataset (**Fig. 3f**). Using this classification, we could further estimate the prevalence of these complex beads to be 6.9% in this dataset (see **Methods**). Parallel analyses for the in-house dataset yielded similar results (83.5% complex bead multiplets; 4.9% heterogeneous beads). Interestingly, the percent difference between the log2 number of valid fragments between these two classes of multiplets showed greater variability for the complex beads than for barcode multiplets presumably caused by two beads (**Fig.**

3g; see **Methods**). This result supports the idea that there may be a predominant individual barcode sequence on these complex beads though there is detectable heterogeneity. Finally, as 10x recently released their v1.1 “NextGem” design, we processed two additional datasets that were run with the two different chip designs in parallel. Our results confirm that the abundance of barcode multiplets persists across both of these two different chip designs (**Fig. 3f**).

This section is supported by the following paragraphs in the methods:

Classifying and quantifying complex beads

To determine multiplets driven by putative bead barcode synthesis errors, we considered all pairs of barcodes within an annotated multiplet and computed the restricted longest common subsequence (rLCS) between them. Explicitly, the rLCS is the largest consecutive number of characters that match between two strings without shifting the strings. We note the necessity of defining a distance metric (rLCS) that is distinguished from the longest common subsequence (LCS) as our metric does not allow insertions or deletions when performing the string matching. Additionally, rLCS is distinguished from the Hamming distance as the matching characters must all occur in a continuous unit (which is not enforced by Hamming).

To determine an appropriate threshold to classify multiplets as having originated from multiple beads or a single heterogeneous bead, we established a null distribution of the rLCS shown in **Fig. S3f**. To achieve this, 1,000,000 random draws of barcode pairs were determined and the rLCS was computed. We selected an rLCS threshold of 6 as pairs with an rLCS ≥ 6 represented less than 0.5% of the data, which was used to classify multiplets from the real data (**Fig. 3f**). To determine whether the number of fragments was similarly captured between barcodes contained in multiplets, we computed the pairwise percent difference of the log₂ unique fragments (“passed_filter” in the CellRanger-ATAC .csv file). The per-multiplet average of the mean pairwise percent difference is plotted in the boxplots in **Fig. 3g**, and we used a two-sided Kolmogorov–Smirnov test to verify that the droplets containing multiple beads had a more even ratio of reads compared to the multiplet driven by bead heterogeneity.

To quantify the percent of beads that had heterogeneity, the numerator was the number of multiplets identified with an rLCS ≥ 6 (from **Fig. 3f**). The denominator was the total number of barcodes analyzed while 1) still counting all barcodes in perceived bead multiplets but 2) collapsing the heterogeneous barcode multiplets to only 1 barcode.

We are grateful to the reviewer for this valuable suggestion.

Reviewer #2 (Remarks to the Author):

10X Chromium has emerged as the leading platform for various types of single cell genomics. At the core of all methods is that one cell and one bead carrying one barcode sequence end up in one reaction droplet, so that all reads resulting from one cell are identifiable by that barcode. Here the authors report, that reads from one cell can be split between two or more barcodes. This could be due to (I) multiple beads per droplet and (II) one bead carrying multiple barcodes. The first option is evaluated using microscopy of the droplets, where the authors find with more than 20% an unusually high fraction of empty droplets and 8% droplets with multiple beads, resulting in 27% of all barcodes to represent cells with barcode multipliers. The second option is evaluated by analysing the Tn5-cut site similarity in scATAC-seq data, using their published software `bap` that was designed to distinguish multiple cells that were caught in the same droplet.

Here the authors turn this logic around and use `bap` to identify cells with unusually similar cut-sites, reasoning that those barcodes mark split cells as they would result from multi-barcode droplets. Using this method, the authors find 13 and 18% of barcode multipliers. As the authors say `bap` was calibrated to find multiple cells / droplet, which is a very different problem in that it aims to find dissimilar cells. The authors did not show anywhere that it works equally well to identify the opposite end of the spectrum, i.e. to identify cells that are unusually similar. Even though these numbers appear to be underestimates, I would still be interested in the estimate of a false positive rate.

We appreciate the reviewer's careful reading of our present work but wish to clarify one important point. In our previous work, multiple cells in single droplets were determined strictly by differences in barcoded Tn5 and not using the cut-site similarity as the reviewer suggested. Indeed, for both scATAC-seq assays, transposition happens before cells are flowed into droplets. Thus, cells co-occurring in the same drop would not have unusually similar cut sites.

Instead, the original logic of `bap` was and is to identify instances of multiple barcodes within a single droplet. In our previous work, this was done deliberately via bead overloading to improve capture efficiency. Here, we sought to find unwanted barcode multipliers in the from the 10x system. However, there is no major conceptual difference-- only an application to this new platform.

Nevertheless, we agree that it is essential to define a well-calibrated false-positive rate in the using 10x data. To achieve this, **Figure 2b-e** now demonstrates an estimation of the false positive rate through mixing two different experiments as suggested below. As we detected no merges between the experiments, we estimate the false positive rate to be low for the analysis of 10x data. This was an excellent suggestion that we hope will reinforce the utility of our method and the validity of our findings. More detail concerning this result is discussed below in the line-item comment.

The paper aims to quantify a technical artifact, however in my opinion neither of the methods was shown to be truly quantitative.

While we agree that there is some uncertainty in our quantification, we emphasize that this report is the first dedicated reporting of this artifact and their confounding in biological interpretations in droplet-based single-cell data, and our estimates and analyses provide an important basis for users to understand and potentially identify these effects in their data. Given that our best efforts to determine a false positive rate have indicated that bap produces very few false-positive merges, our approach may be currently underestimating the severity of the artifact in single-cell data, but we find that the current rates, even if underestimated, warrant publication to make users broadly aware of this phenomenon. We expect that future work can build on our substantial contribution to more precisely quantify this artifact.

As far as I understood the authors tested only one run from one 10X machine under standard settings for the number of beads/droplet. In order to generalise this conclusion, I would like to see replicates, maybe even on a different machine, to catch the full technical variance. This is of particular importance because the authors already report unusual behavior of their current results with respect to the fraction of empty beads. It is necessary to exclude the possibility that their observation is due to a badly calibrated 10X machine.

We thank the reviewer for this important suggestion. Notably, we point to the reviewer that our analysis of this counting issue comes from data gathered from various sources, including data generated by us, gold-standard data provided by 10x, and data from single-cell RNA-seq experiments. We also note that much of our confounding barcodes arise from heterogeneous beads (rather than multiple beads per droplet). We apologize that this was not more clear in the initial submission, we have further discussed the point and also included an additional quantitation disambiguating cases where we expect multiple beads versus heterogeneous beads (see response X).

Further, following this recommendation, to more precisely address this concern with the imaging results we have performed two additional imaging experiments. In particular, we performed another technical replicate (same machine; same training kit) as well as another experiment using the v1.1 kit on a different machine at a different research institute. Additionally, we depict our data in the context of the published results from Zheng *et al.* 2017 to provide a more comprehensive look at the potential for bead doublets to occur in droplets. The new **Fig. 1c,d** summarize these results as shown below:

Figure: (c) Empirical quantification of number of bead barcodes based on image analysis over 3 replicates with previously published data (Zheng *et al.* 2017²). (d) Percent of barcodes associated with multipliers under the distribution observed in (c). Error bars represent standard error of mean over the experimental replicates. Panels were taken from Figure 1.

Moreover, we highlight **Fig. S1f** that shows the results of each individual replicate imaging experiment.

Figure: Quantification of beads per droplet for each replicate. Above each panel, the machine and the version of the chip used for the training kit is indicated. Error bars represent standard error of mean over 10 fields of view per replicate. Panel from Fig. S1f.

Overall, our averages shown in **Figure 1** now closely match what has been previously reported in Zheng *et al.* 2017. However, as the results vary considerably, we emphasized this point in the revised text:

On average, we found that 16.1% of droplets contained no beads, 80.0% with exactly one bead, and 3.9% having two or more beads (**Fig. 1c**). These results were consistent with the previously reported results of this platform (“Zheng”)² and confirm the

sub-Poisson loading of beads into droplets (compare to **Fig. S1e** for optimal Poisson loading). While the mean of the bead loading was consistent with previous reports, we note considerable run-to-run variability from our imaging replicates, ranging from 0.8% to 8.4% (**Fig. S1f**). These results indicate that the occurrence of bead multiplets likely varies between machines and individual runs. Furthermore, we noted occurrences of large droplets with multiple beads (**Fig. S1g**) that likely originated from the errant merging of several individual droplets, yielding another source of potential barcode multiplets.

While our estimate of the occurrence of multiple beads in droplets confirms previous reports², we emphasize that this problem exacerbated when considering potential barcodes in single-cell data. On average, we estimate that 11.4% of barcodes would represent barcode multiplets, reflecting droplets with heterogeneous oligonucleotide sequences (**Fig. 1d**; see **Methods**).

The entire second part is based on the premise that barcodes from the same droplet can be faithfully identified using their previously published method bap. To begin with, the method needs a more detailed description and evaluation:

What exactly went into the calculation of the Jaccard index (Coverage filters ...)?

To better facilitate an explicit description and understanding of our approach using 10X single-cell ATAC-seq data, we've included a supplemental figure (**Fig. S2; shown below**) that outlines the computational workflow. Specifically, no coverage filters are needed as we used the high-quality, error-corrected barcodes that are provided from the CellRanger-ATAC preprocessing pipeline. Using the combination of pre-identified barcodes and the raw sequencing reads for the experiment (both outputs from a default execution of CellRanger-ATAC), bap then computes the jaccard index after 1) filtering for pre-specified barcodes and 2) removing fragments present over-represented across many barcodes (by default, we filter fragments present in more than 6 barcodes but this customizable using the `-nc` flag; further we note the default of 6 is motivated in our previous work). Afterwards, the jaccard index is computed. These steps are summarized graphically in the supplemental figure.

Figure: (see Fig. S2). An overview of the inputs and computational workflow for the application of bap to 10x scATAC-seq data.

How does the metric depend on coverage?

This is an excellent question. To directly address this, we've included a new figure panel that compares which pairs of barcodes are identified at various abundances of down-sampling. After nominating a "true-positive" set of pairs at 100% sequencing depth, we performed downsampling at intervals of 10% to determine the proportion of merged pairs identified. As indicated in the new supplemental panel shown below, we suggest that at a depth of 10,000 fragments detected per cell, the metric from bap becomes very stable. Importantly, at each of the intervals, **no barcode pairs were identified above the knee that were not present at 100% sequencing depth (false positives)**, indicating a high specificity of the metric. In summary, the down-sampling analysis demonstrates low false-negatives to approximately 40% of the sequencing depth, and no identified false-positives as a function of downsampling.

We note the inclusion of the following sentence to the main text:

Additionally, through iteratively downsampling and re-executing bap, we confirmed the stability of our metric with sequencing depths as low as a median 10,000 fragments detected per barcode (**Fig. S3e**; see **Methods**), confirming the broad utility of this approach.

And the following paragraph to the methods section:

To evaluate the stability of the bap statistic as a function of coverage, we downsampled the dataset generated here ("this study") at intervals of 10% and reran bap on the resulting downsampled .bam files. Here, we used the full set of high-quality barcodes determined from the CellRanger execution on the full dataset. Moreover, we determined the set of identified barcode pairs from the full dataset as a 'true positive' set of pairs to compare the downsampled results. **Fig. S3e** shows the results of this downsampling, including the 40% subsample (that corresponded to a median 10,132 fragments per barcode) that achieved >90% sensitivity in detecting the set of barcode pairs from the full data. Critically, in each of the 9 downsampled executions of bap, no barcode pairs were identified that were not present in the full dataset.

Could the multiple barcodes per cell be due to sequencing errors?

We do expect sequencing errors to contribute to the issue of multiple barcodes. However, our inferences indicate that sequencing error would only represent a small fraction of what we detect here. Further, in our analyses we developed an analytical approach so that we could be agnostic to the source of error.

Specifically, the inputs of bap (which we've clarified in **Figure S2** in the revised version shown below) are processed .bam files and high-quality barcode files produced by the default execution of CellRanger-ATAC. Importantly, CellRanger-ATAC corrects for sequencing errors by assigning fragments to the closest valid barcode of the ~737,000 valid barcodes for the 10x assays. Additionally, any single barcode is separated by a Hamming distance of at least 3, which provides additional robustness in assigning fragments to barcodes. By using the CellRanger-ATAC output, which already explicitly corrects for potential sequencing errors and uses a predefined set of high-quality barcodes, we find it highly unlikely that sequencing errors explains the multiple barcodes.

Figure: (see Fig. S2). An overview of the inputs and computational workflow for the application of bap to 10x scATAC-seq data.

To further clarify this point, we've added the following sentence to the main text:

Additionally, since individual barcodes in the space of all possible barcodes are separated by a minimum Hamming distance of three in the 10x platform, these the high prevalence of barcode multiplets are unlikely to be caused by sequencing errors.

What is the Hamming distance between the collapsed barcode pairs?

We thank the reviewer for this valuable suggestion to investigate this. First, we acknowledge a revised paragraph in the Methods section that motivates the use of the rLCS metric over the Hamming distance:

Classifying and quantifying complex beads

To determine multiplets driven by putative bead barcode synthesis errors, we considered all pairs of barcodes within an annotated multiplet and computed the restricted longest common subsequence (rLCS) between them. Explicitly, the rLCS is the largest consecutive number of characters that match between two strings without shifting the

strings. We note the necessity of defining a distance metric (rLCS) that is distinguished from the longest common subsequence (LCS) as our metric does not allow insertions or deletions when performing the string matching. Additionally, rLCS is distinguished from the Hamming distance as the matching characters must all occur in a continuous unit (which is not enforced by Hamming).

Next, we indicate the distribution below (also shown in **Fig. 3f**).

Figure: (see Fig. 3f) Empirical distribution of the mean restricted longest common subsequence (rLCS) per multiplet. A cutoff of 6 was used to determine either of the two classes of barcode multiplets.

As discussed at various other points in this response, the functional difference in the distance between barcode pairs was extremely useful in estimating other components associated with barcode multiplets.

What are the ratios of reads for the split barcodes? For the 27% of split-cells that are due to multiple beads, I would expect a more even ratio than for mixed barcodes on the same bead.

We appreciate the reviewer for asking this extremely interesting question. Using the classification of multiplets having likely originated from multiple beads or complex beads (left panel below; also **Fig. 3f**), we answered this question and confirmed the reviewer's expectations (shown in **Fig. 3g**).

Figure: (left; see Fig. 3f) Empirical distribution of the mean restricted longest common subsequence (rLCS) per multiplet. A cutoff of 6 was used to determine either of the two classes of barcode multiplets. (right; see Fig. 3g) Percent difference of the mean log₂ fragments between pairs of barcodes with in a multiplet. The reported p-value is from a two-sided Kolmogorov–Smirnov test.

We also highlight the relevant addition to the Methods section:

To determine an appropriate threshold to classify multiplets as having originated from multiple beads or a single heterogeneous bead, we established a null distribution of the rLCS shown in **Fig. S3f**. To achieve this, 1,000,000 random draws of barcode pairs were determined and the rLCS was computed. We selected an rLCS threshold of 6 as pairs with an rLCS ≥ 6 represented less than 0.5% of the data, which was used to classify multiplets from the real data (**Fig. 3f**). To determine whether the number of fragments was similarly captured between barcodes contained in multiplets, we computed the pairwise percent difference of the log₂ unique fragments (“passed_filter” in the CellRanger-ATAC .csv file). The per-multiplet average of the mean pairwise percent difference is plotted in the boxplots in **Fig. 3g**, and we used a two-sided Kolmogorov–Smirnov test to verify that the droplets containing multiple beads had a more even ratio of reads compared to the multiplet driven by bead heterogeneity.

What is the false positive rate for the bap barcode merging? The authors could analyse technical replicates from 2 different 10X runs to explore how often the Jaccard index would suggest to merge cell barcodes from two distinct runs.

We thank the reviewer for the suggestion to assess the false positive rate. Indeed, we conducted this exact experiment as described in the new methods section below:

In silico mixing experiment

Using two different public PBMC 5k datasets, we sought to determine a putative false positive rate for the application of bap to 10x scATAC-seq data. Here, we denoted the PBMC-5k “Public” dataset as Channel 1 and the PBMC-5k from the NextGEM beads as

Channel 2. We modified the CB tags (which contains the error-corrected barcodes) in the .bam files for each channel to ensure that each barcode for each experiment was uniquely identifiable. These modified bam files were subsequently merged. Next, the same modification to the barcodes was made, and the two high-quality barcodes files were combined into a single file. We then executed bap using the default parameters with this merged .bam and merged barcode list file. Using a single threshold determined by the knee call, we identified pairs of barcodes originating from the same or different channels as summarized in **Fig. 2c-e**. The top 500,000 barcode pairs were plotted in rank order for each of these three plots, and the same single threshold was visualized in all three panels.

The results of this mixing experiment now provides an important conceptual validation of the use of bap in the 10x dataset, which now comprises most of **Fig. 2** shown below:

Figure (see Figure 2): (b) Schematic of mixing experiment. Two channels were combined and the resulting merged files were analyzed with bap. (c-e) Knee plots comparing the top 500,000 barcode pairs from (c) only channel 1, (d) only channel 2, and (e) between channels. The number of pairs calls is indicated by the number of points above the blue horizontal line (see Methods).

We highlight the corresponding edit in the text:

After updating bap to facilitate processing of the 10x scATAC data (**Fig. S2**; see **Methods**), we conducted an initial *in silico* experiment in order to verify the applicability of our approach to 10x scATAC-seq data. Here, we combined two channels from a similar biological source (~5,000 cells of peripheral blood mononuclear cells; PBMCs) and executed bap on the resulting combination (**Fig. 2b**; see **Methods**). As any barcode pairs merged between channels would be false positives, our approach facilitated an estimation of the false positive rate of our approach in 10x data. After executing bap with the default parameters, 1,874 barcode pairs were identified as sharing an unusual number of shared transposition events. Specifically, 931 pairs from channel 1 (**Fig. 2c**) and 943 pairs from channel 2 (**Fig. 2d**) were identified. However, zero pairs were identified between channels (**Fig. 2e**), indicating a very low false positive rate for bap when applied to this assay. Moreover, the shape of the ranked-ordered barcode pair curves for the channels separately were distinct from the between-channel curve (**Fig.**

2c-e). Overall, these results support the utility of bap in inferring barcode multiplets from the 10x platform.

We thank the reviewer for this excellent suggestion.

If part of the split-cells can be identified by a simpler measures such as the Hamming-distance between barcodes, maybe in combination with a frequency check, similar to the suggested algorithm in UMI-tools, the generalisation from ATAC-seq to other data would be more convincing.

We agree that establishing the generalization of split-cell identification using barcode similarity would strengthen the manuscript. To achieve this, we've included **Figure S3f** (shown to the right) that indicates the distribution of barcode distance (using rLCS rather than Hamming as a slightly more appropriate measure) for 1,000,000 random barcode pairs. This effectively creates a “null” distribution where barcode pairs with an rLCS ≥ 6 would have a 0.5% chance of co-occurring assuming an independent assignment of barcodes to a particular cell. As the generalizations to scRNA-seq (now in **Figure 4**) show rLCS values ≥ 9 ($< 0.01\%$ in a null model), we believe that this analysis provides a convincing logic for the transferred application. Finally, we note that the barcode sequences underlying the beads for the ATAC and RNA-seq technologies are the same which provides additional validation for our approach.

While the reviewer’s suggestion to combine the barcode similarity metric with a frequency check, we suggest that this would require the development of a new algorithm that extends beyond the scope of the current study. Moreover, we envision challenges in validating this approach in clonal cell types (especially those marked by BCRs and TCRs that would have very similar transcriptomes) where there may be a limit in the overall molecular diversity.

The null distribution of barcode pairs is emphasized in this new part of the main text:

To achieve this, we established a null distribution by computing the rLCS for random pairs of barcodes from the 10x whitelist (see **Methods**). Over 1,000,000 sampled pairs, we determined that pairs with an rLCS ≥ 6 were extremely uncommon assuming an independent co-occurrence ($< 0.5\%$ probability of co-occurring; **Fig. S3f**). Thus, for inferred multiplets with a mean rLCS ≥ 6 , we interpret these to be most likely caused by heterogeneous barcodes within a single bead.

Furthermore, based on this suggestion, we note the additional text now present in the discussion that includes this point:

We envision a combination of dense exogenous barcodes via cell hashing¹⁰ and evolved by CRISPR-Cas9¹¹ or intrinsic features such as clonal mutations, rearrangements, or highly correlated abundances with barcode sequence similarity metrics could be leveraged to better infer barcode multiplets.

I am missing a lot of information pertaining the experimental details, i.e. which 10X chemistry was used? Did this differ between the public and their own dataset etc ... This is of utmost important for this paper because it describes a technical artifact, therefore the scope this analysis critically depends on such technical details.

We apologize for the brevity of information regarding the experimental details. Currently, only one chemistry has been made available for the 10x Chromium scATAC-seq solution. We have now made this more explicit in the methods in the revised version. Additionally, we note that in the past few months, 10X has released the NextGEM 1.1 chips (which can be used in all assays, including scRNA-seq and scATAC-seq). While the chemistry of the ATAC-seq reaction is the same, we did evaluate the effects the NextGEM 1.1 chip comparing additional public data. Specifically, as part of the new public datasets, 10x has released head-to-head comparisons comparing the chip versions. For the same input cell experiment (pbmc-5k), we computed the barcode multiplet rates using bap (two new datasets than those previously analyzed). These analyses confirmed that the artifact persisted at comparable levels to our previous analysis, which is summarized in the panel to the right as well as in **Fig. 3h**.

We highlight the inclusion of these analyses in the main text:

Finally, as 10x recently released their v1.1 “NextGem” design, we processed two additional datasets that were run with the two different chip designs in parallel. Our results confirm that the abundance of barcode multiplets persists across both of these two different chip designs (**Fig. 3f**).

And in the methods:

For the comparison of the chip technologies (**Fig. 3g**), we again downloaded the PBMC 5k scATAC-seq datasets from the “Chromium Next GEM ATAC Demonstration.”

and

The cells were then pelleted again, resuspended in 1 ml PBS, and used for 10x ATAC v1.0 protocol following manufacturer's instructions.

References

Zheng, G. X. Y. *et al.* Massively parallel digital transcriptional profiling of single cells. *Nat. Commun.* **8**, 14049 (2017).

Reviewers' Comments:

Reviewer #1:

Remarks to the Author:

The revised manuscript has substantially improved over the initial submission by adding technical replicates for imaging analysis, simulation-based assessment of the multiplets calling method, quantification of the contribution of the two sources of multiplets. However, there are a still few points that need addressing.

1) The number of technical replicates for the imaging analysis is too small to support claims about technical variability. Linked to this issue, it is hard to tell whether or not the 1st droplet imaging run of this study, seemingly quite different from the other runs and driving the large variance, genuinely reflects technical variability intrinsic to the 10x platform. With the contribution of this run, the authors claimed that 3.9% of droplets are multi-bead droplets. Later, the authors showed that multi-beads account for only ~15% of barcode multiplets while estimating 4.9-6.9% of beads are complex beads. These numbers seem to contradict each other, which perhaps suggests the estimated multi-beads proportion is too high.

2) The method for clone size correction (fig 4 d,e) seems to simulate droplets rather than barcodes ("To achieve this, we sample from M' until the sum meets or exceeds cj. cj' then is the number of draws corresponding to the number of multiplet-aware droplets needed to explain the clonotype abundance and can be interpreted as the number of cells present in the clone under the simulation setting."). Therefore, the proportion of simulated barcodes affected by multiplets is $(2*0.1+3*0.02+4*0.02+5*0.01)/(1*0.85+2*0.1+3*0.02+4*0.02+5*0.01) = 0.315$, which greatly exceeds the reported proportions (13.2% or 17.6%) of the two examined datasets from which the simulated multiplicity distribution was derived. Consequently, the degree clonal inference is affected by multiplets is likely over-estimated.

3) The occurrence of barcode synthesis error is quite unlikely an on/off process but might exist at a spectrum of frequencies for all beads. Due to lower starting copy number of templates in each cell/droplet in scATACseq compared to scRNAseq, rare oligo barcodes with synthesis errors can have a higher chance of drifting to sufficiently high frequencies that evade dropout or filtering. Therefore, the observed complex-bead rate might be assay-dependent and extrapolating observed multiplicity distribution from scATACseq to scRNAseq might be problematic. This is not to discount the authors discovery of this source of error, but the point is worth discussing. Also, given that the authors already observed unexpectedly higher frequency of long rLCS in inferred TCR/BCR clones (fig S4 a), why not use that to estimate the complex-bead rate in scRNAseq, and, combined with multi-bead rate estimated from imaging analysis, derive the multiplicity distribution for simulation in the clone size correction analysis?

4) The inclusion of nextGem data is useful but could be more informative if more data, particularly contribution of the two sources of multiplets, is presented at least in the supplementary, as it seems that a major difference introduced by nextGem is the new flowcell design which might affect multi-bead rate.

Overall, the main conclusion of the manuscript is convincing and of great interest to the field, based on which we would recommend its publication after appropriate revision.

.

Reviewed by Ni Huang, Carlos Talavera-López and Sarah A. Teichmann - Wellcome Sanger Institute.

Reviewer #2:

Remarks to the Author:

I thank the authors for carefully addressing all my concerns and suggestions.

Reviewer #1 (Remarks to the Author):

The revised manuscript has substantially improved over the initial submission by adding technical replicates for imaging analysis, simulation-based assessment of the multiplets calling method, quantification of the contribution of the two sources of multiplets. However, there are still a few points that need addressing.

1) The number of technical replicates for the imaging analysis is too small to support claims about technical variability. Linked to this issue, it is hard to tell whether or not the 1st droplet imaging run of this study, seemingly quite different from the other runs and driving the large variance, genuinely reflects technical variability intrinsic to the 10x platform. With the contribution of this run, the authors claimed that 3.9% of droplets are multi-bead droplets. Later, the authors showed that multi-beads account for only ~15% of barcode multiplets while estimating 4.9-6.9% of beads are complex beads. These numbers seem to contradict each other, which perhaps suggests the estimated multi-beads proportion is too high.

We appreciate the careful consideration regarding this point. After informal conversations with colleagues at 10X, we were informed that the products used in the test kits (which are necessary for these imaging experiments) undergo less rigorous quality-control assessments than beads used in a typical run. As noted by 10x, our observations surrounding the technical variability may not fully reflect those seen in single-cell data (in addition to the low number of experimental replicates as the reviewer suggests). In response to these comments, we've now modified statements in the revised version of the text to raise awareness of this potential issue, we also emphasize that imaging is an imperfect approach for measuring the multi-bead artifact in single-cell data, the results nonetheless warrant attention of users of these technologies, which we've now clarified in the text:

While our imaging results indicate that the occurrence of bead multiplets likely varies between machines and individual runs, we note that the training kits are only a proxy for the reagents used in producing single-cell data, and may reflect a higher rate of bead doublets. Though imperfect, our results suggest that multiple beads may co-occur in droplets and motivates additional computational analysis to determine potential barcode multiplets. (Page 3)

Rather than performing additional rounds of imaging that may ultimately capture effects that do not directly align with the the true single-cell data, we hope that our revised statement more appropriately contextualizes these results.

In terms of the rates potentially being at odds with each other, we emphasize that the mean of our imaging runs matches estimates from 10x. Furthermore, we report the complex bead rate for two additional replicates (see point 4; **Fig. S3g**) that agree with our previously reported estimates, and our evaluation of the 10x software solution further confirms the estimates reported from bap (see last point). Thus, we believe that each of these numbers individually are reasonably accurate. Ultimately, as the results come from different sources (multi-bead from imaging; complex bead from sequencing) and require assumptions that may not , discretion must be employed when comparing these rates, a discussion which we now communicate in the text:

As the rates inferred from imaging and from bap are derived from distinct sources (i.e. bead/droplet counting versus sequencing), discretion is required when comparing between the detection modalities. (Page 7)

2) The method for clone size correction (fig 4 d,e) seems to simulate droplets rather than barcodes ("To achieve this, we sample from M' until the sum meets or exceeds c_j . c_j ' then is the number of draws corresponding to the number of multiplet-aware droplets needed to explain the clonotype abundance and can be interpreted as the number of cells present in the clone under the simulation setting."). Therefore, the proportion of simulated barcodes affected by multiplets is $(2*0.1+3*0.02+4*0.02+5*0.01)/(1*0.85+2*0.1+3*0.02+4*0.02+5*0.01) = 0.315$, which greatly exceeds the reported proportions (13.2% or 17.6%) of the two examined datasets from which the simulated multiplicity distribution was derived. Consequently, the degree clonal inference is affected by multiplets is likely over-estimated.

We apologize for the confusion regarding this point and appreciate the careful examination. While we had defined m_i to be a barcode-level statistic, we agree that the sampling procedure effectively simulates droplets, leading to an overestimated degree of clonal inference in our previous implementation. In our revised version, we've now re-parameterized as follows:

$$P(m_i = 1) = 0.93; P(m_i = 2) = 0.05; P(m_i = 3) = 0.01; P(m_i = 4) = 0.005; P(m_i = 5) = 0.005$$

Using the analogous computation to what the reviewer's perform, this results in a 15.8% barcode multiplet rate, in line with values reported in our manuscript. We've also modified text where appropriate to better communicate this result:

Overall, the percentage of cells associated with a clonotype comprised of at least two cells decreases considerably for both BCR (24.5% to 18.6%; Fig. 4d) and TCR (23.6% to 17.9%; Fig. 4e) clonotypes. Further analyses indicated a clone false discovery rate as high as 23.5% (BCR) and 22.5% (TCR) in these data (Page 7)

Further, we've added a new **Table S4** that provides analogous values in our true datasets to verify that our values for the simulation are appropriate.

We appreciate the close and thoughtful reading of our methods to identify this point.

3) The occurrence of barcode synthesis error is quite unlikely an on/off process but might exist at a spectrum of frequencies for all beads. Due to lower starting copy number of templates in each cell/droplet in scATACseq compared to scRNAseq, rare oligo barcodes with synthesis errors can have a higher chance of drifting to sufficiently high frequencies that evade dropout or filtering. Therefore, the observed complex-bead rate might be assay-dependent and extrapolating observed multiplicity distribution from scATACseq to scRNAseq might be problematic. This is not to discount the authors discovery of this source of error, but the point is worth discussing.

We completely agree with this comment and have added the following paragraph to the discussion:

Conceptually, the presence of heterogeneity in beads is unlikely to be caused by an on/off process and instead likely exists as a spectrum across all beads used in these assays. As the estimated number of complex beads relies on sufficient amplification and detection of lower-frequency barcodes inside of droplets, the proportion of barcodes affected by this artifact becomes a function of the read depth (Fig. S3e) and the barcode threshold (Fig. S3b), which are in turn functions of the underlying chemistry of the assays. While our estimation of the clone false

discovery rate assumed comparable rates for barcode multiplets for scATAC-seq and scRNA-seq methods, technical differences across these assays could also result variable barcode multiplet abundances. As such, our work motivates further investigation into the relationship between barcode multiplets and clonal diversity across various technical platforms.

Importantly, we think that emphasizing the putative spectra rather than binary nature of the multiplets provides an important conceptual point for the readers, and we thank the reviewer for suggesting this idea.

Also, given that the authors already observed unexpectedly higher frequency of long rLCS in inferred TCR/BCR clones (fig S4 a), why not use that to estimate the complex-bead rate in scRNAseq, and, combined with multi-bead rate estimated from imaging analysis, derive the multiplicity distribution for simulation in the clone size correction analysis?

This is an interesting suggestion. After some investigation and thought, we believe that this idea is complicated by the fact that we cannot accurately estimate the complex-bead rate in scRNA-seq. For the B-cell example, we identified putative complex beads by identifying groups of barcodes that 1) shared a clonotype receptor and 2) had a rLCS ≥ 6 (identical to the procedure for ATAC). Of the 680 barcodes considered, we identified 570 unique “groups”, including 88 that had multiple barcodes, suggestive of a 15% (from this computation: $(88/570 * 100\%)$) complex bead rate, which is $\sim 3x$ greater than we estimated in the ATAC data though the calculation was essentially the same.

We anticipate that this 15% number is inflated as bap provides a higher burden of evidence for pairs of barcodes belonging to a multiplet than co-occurrence in a clonotype. In particular, though a rLCS ≥ 6 is a rare phenomenon by random chance ($< 0.5\%$ by permutations), clones contain as many as 98 barcodes, leading to thousands of pairwise combinations and thus substantial potential false positives for the complex bead estimation. While we could in theory further modify our simulation framework to accommodate this complication, we ultimately have to rely on the (likely imperfect) imaging to estimate the multiple bead rate, as the reviewers suggest, which would further inflate the of barcode multiplets. Taken together, the putative “clone false discovery rate” would be much higher if we attempted to derive it from the RNA-seq data alone, and here, we’ve chosen to report the more conservative estimate, which we make clear to the readers in the revised text:

Finally, we note that while this simulation assumes that the multiplet rates inferred for scATAC-seq are transferable to scRNA-seq, alternative approaches, such as estimating the complex bead rate from scRNA-seq directly, are likely unreliable without a sensitive multiplet detection approach as presented with bap. Ultimately, we believe our results from this simulation to be relatively conservative and provide a reliable anchor to interpret clonotype abundance in the lense of our barcode multiplet artifact. However, additional experiments and analytical tools are needed to accurately determine clonotype abundance. (Page 25)

Ultimately, after the new simulation parameterization that has been carried out (as suggested in point 2), we believe the newly reported results represent a conservative yet alarming rate that most appropriately communicates this concern to the single-cell community. Importantly, the presumed clone false discovery rate is now reported as 23.5% for the BCR clones and 22.5% for the TCR clones, which has been updated in the text (see Page 7).

4) The inclusion of nextGem data is useful but could be more informative if more data, particularly contribution of the two sources of multipliants, is presented at least in the supplementary, as it seems that a major difference introduced by nextGem is the new flowcell design which might affect multi-bead rate.

We agree with this point and have now added a new supplemental panel that shows these rates for the two additional datasets side-by-side (**Fig. S3g**). Overall, the rates are quite similar to what we reported for the other datasets. Though we agree the new design of the NextGem could have lead to major differences, we do not see evidence of this in the data. (Panel legend: Breakdown of types of barcode multipliants from the Next-gem comparison data.)

Overall, the main conclusion of the manuscript is convincing and of great interest to the field, based on which we would recommend its publication after appropriate revision.

We appreciate the positive outlook and very useful comments regarding our work. During the course of the last revision, we note that 10x has released a community letter confirming our findings in the pre-print version of this manuscript and is now updating their software solution in response to our findings: <https://community.10xgenomics.com/t5/10x-Blog/Letter-from-10x-Genomics/ba-p/68555>. To further enable the single-cell community, we have evaluated their software solution and have included this comparison in the methods:

After contacting 10x support, we obtained the “clean_barcode_multipliants_1.0.py” script, which identifies barcode multipliants in single-cell ATAC-seq data. We executed this code and evaluated the output for the two scATAC-seq datasets closely analyzed in this work (“Public” and “This Study”). While the procedure used to identify multipliants similarly utilizes shared Tn5 insertions, the treatment of multipliants once detected is different from bap. Specifically, for each multipliant, the barcode with the most unique fragments is retained and the other barcodes are filtered out. Further, 10x refers only to the barcodes that are filtered out as ‘multipliants’, rather than counting the most prevalent barcode as part of a barcode multipliant as we’ve done throughout this manuscript. For comparison purposes, we used our definition of barcode multipliant (as stated in the abstract) and reported the rates from each tool (see script in Code Availability for the exact procedure). Finally, to compute the concordance between the two methods, we assigned each barcode whether or not it was part of a barcode multipliant from both sources and report the percentage of barcodes that had a matching annotation across the detection methods (Page 24).

Importantly, the concordance of the two methods (10x solution and bap) was >98%, and the 10x solution actually reported a slightly higher absolute rate of barcode multipliants, as we show in **Fig. S3h** (also shown below):

Panel: (see Fig. S3h) Comparison of methods to detect barcode multiplets. The rates of barcode multiplets detected by each solution is shown in black. The % agreement between the two methods (per barcode) is shown in red.

These additional results are now summarized in the main text:

In response to a pre-print version of our article⁶, 10x Genomics released a letter a software solution to identify multiplets from the output of the CellRanger-ATAC pipeline. In principle, their approach similarly utilizes the molecular diversity of Tn5 cut sites to identify putative barcode multiplets. After obtaining this script, we evaluated our two well-characterized PBMC datasets and determined that the rates of barcode multiplets were extremely similar as >98% of barcodes were concordantly classified as belonging to a barcode multiplet or not (Fig. S3h; see Methods). As a solution to the barcode multiplet artifact, the 10x method discards the lower abundance barcodes per multiplet. While further analysis is required to determine the optimal strategy for handling barcode multiplets, these results corroborate our estimates inferred and reported from bap. (Page 6)

Reviewer #2 (Remarks to the Author):

I thank the authors for carefully addressing all my concerns and suggestions.

We appreciate the thoughtful suggestions in the first round of revision and appreciate the positive feedback on our revised version of the manuscript.

Reviewers' Comments:

Reviewer #1:

Remarks to the Author:

I thank the authors for the thorough response which addressed all except one of our concerns.

Regarding the point of estimating complex bead rate from scRNAseq data using rLCS within B-cell clones, I completely agree that sharing the same tn5 fragments is a stronger evidence and relying solely on rLCS can increase false positive rate. Given the authors' calculation, the 680 barcodes considered and the 570 groups used as denominator seem to have excluded one-cell clones.

Can the authors clarify if the 4.9-6.9% complex bead rate estimates from ATACseq data have also excluded singletons? If yes, what is the rationale behind it? If not, then following the same methods, the rate would be $88/(570+2095)=3.3\%$.

Further taken into account the chances of random assortment of barcodes (potentially multi beads) shown in figS4a, the complex bead rate would be even lower. Therefore, I am quite unsure about the sentence "Ultimately, we believe our results from this simulation to be relatively conservative and provide a reliable anchor to interpret clonotype abundance in the lense of our barcode multiplet artifact." If my calculation above is correct, that even with increased false positives the complex bead rate is lower in scRNAseq than in scATACseq, which also fits the hypothesis that synthesis error exists in a spectrum across all beads and lower copy number of input can amplify observed rate.

Regarding the point of estimating complex bead rate from scRNAseq data using rLCS within B-cell clones, I completely agree that sharing the same tn5 fragments is a stronger evidence and relying solely on rLCS can increase false positive rate. Given the authors' calculation, the 680 barcodes considered and the 570 groups used as denominator seem to have excluded one-cell clones.

This is correct-- the 680 barcodes were from the set of 2,775 total barcodes, resulting in 2,095 barcodes (as the reviewers use below) were from singleton clones. We apologize for this exclusion. However, we emphasize that this figure was not reported in the text, only the response to the reviewer.

Can the authors clarify if the 4.9-6.9% complex bead rate estimates from ATACseq data have also excluded singletons? If yes, what is the rationale behind it? If not, then following the same methods, the rate would be $88/(570+2095)=3.3\%$.

We did not exclude multiplets from the scATAC-seq computation and apologize for the confusion. Here, we outline how this rate is computed explicitly for the "This Study" dataset.

Total barcodes: 5,453

- 4,732 barcodes from singlets
- 121 barcodes with multiplet beads per droplet (and thus not complex)
- 600 barcodes in 253 complex beads

$$\text{complex bead rate} = \frac{\# \text{ complex beads}}{\# \text{ singlet beads} + \# \text{ beads in bead multiplets} + \# \text{ complex beads}} = \frac{253}{4732 + 121 + 253} = 4.95\%$$

Thus, the 4,732 term shows that singletons are included in these rate estimations.

To clarify this point, we've added this example calculation on pages 22-23 of the methods section:

For example, in the "This Study" dataset, the total number of barcodes passing the CellRanger knee was 5,453. Of these, 4,732 barcodes were from singlets, 121 barcodes were associated with multiplet beads per droplet (and thus not complex), and 600 barcodes were associated with 253 complex beads. The complex bead rate can be computed as follows:

$$\text{complex bead rate} = \frac{\# \text{ complex beads}}{\# \text{ singlet beads} + \# \text{ beads in bead multiplets} + \# \text{ complex beads}}$$

For our example of the "This Study" dataset:

$$\frac{253}{4732 + 121 + 253} = 4.95\%$$

Further taken into account the chances of random assortment of barcodes (potentially multi beads) shown in figS4a, the complex bead rate would be even lower. Therefore, I

am quite unsure about the sentence "Ultimately, we believe our results from this simulation to be relatively conservative and provide a reliable anchor to interpret clonotype abundance in the lense of our barcode multiplet artifact." If my calculation above is correct, that even with increased false positives, the complex bead rate is lower in scRNAseq than in scATACseq, which also fits the hypothesis that synthesis error exists in a spectrum across all beads and lower copy number of input can amplify observed rate.

We agree that the spectrum hypothesis makes it challenging to definitively interpret our results as conservative estimates. We emphasize that our previous version of the article included a paragraph dedicated to this idea, including the following sentence:

While our estimation of the clone false discovery rate assumed comparable rates for barcode multiplets for scATAC-seq and scRNA-seq methods, technical differences across these assays could also result variable barcode multiplet abundances. (Page 7)

To address the sentence of concern, we have removed it from the revised version of the text, replacing it with the following:

Ultimately, our simulation results provide an anchor to interpret the potential shift in clonotype abundance from the lense of our barcode multiplet artifact. (Page 25)

We thank the reviewers for their careful consideration of this phenomenon and our reported rates.

Reviewers' Comments:

Reviewer #1:

Remarks to the Author:

We thank the author for their very fast addressing of the issues raised.

After reviewing their responses and re-reading the new version of the manuscript, we are satisfied with their responses and recommend this article for publication.